# Structural basis of α-latrotoxin transition to a cation-selective pore

B. U. Klink [1,2,4], A. Alavizargar [2,3,4], K. S. Kalyankumar [1,2], M. Chen[1,2], A. Heuer [2,3] ✉ & C. Gatsogiannis [1,2] ✉

The potent neurotoxic venom of the black widow spider contains a cocktail of seven phylum-specific latrotoxins (LTXs), but only one, α-LTX, targets vertebrates. This 130 kDa toxin binds to receptors at presynaptic nerve terminals and triggers a massive release of neurotransmitters. It is widely accepted that LTXs tetramerize and insert into the presynaptic membrane, thereby forming $Ca^{2+}$-conductive pores, but the underlying mechanism remains poorly understood. LTXs are homologous and consist of an N-terminal region with three distinct domains, along with a C-terminal domain containing up to 22 consecutive ankyrin repeats. Here we report cryoEM structures of the vertebrate-specific α-LTX tetramer in its prepore and pore state. Our structures, in combination with AlphaFold2-based structural modeling and molecular dynamics simulations, reveal dramatic conformational changes in the N-terminal region of the complex. Four distinct helical bundles rearrange and together form a highly stable, 15 nm long, cation-impermeable coiled-coil stalk. This stalk, in turn, positions an N-terminal pair of helices within the membrane, thereby enabling the assembly of a cation-permeable channel. Taken together, these data give insight into a unique mechanism for membrane insertion and channel formation, characteristic of the LTX family, and provide the necessary framework for advancing novel therapeutics and biotechnological applications.

Latrotoxins (LTXs) are the main toxic components of the venom of black widow spiders (*Latrodectus*)[1]. The venom includes the vertebrate-specific α-latrotoxin (α-LTX)[2,3], five insecticidal toxins known as α, β, γ, δ, and ε-latroinsectotoxins (LITs)[4–6], as well as a toxin specific to crustaceans named α-latrocrustatoxin (α-LCT)[7]. Upon envenomation, LTXs impact the victim's nervous system by triggering massive neurotransmitter release upon binding to receptors[8–12]. α-LTX has been widely used as a molecular tool to study the exocytosis of synaptic vesicles and its actions are considered precisely the opposite of those of botulinum and tetanus toxins, both of which inhibit instead of activating the same secretory apparatus[13].

LTXs are large proteins, ranging from 110 kDa to 140 kDa, and they share a common architecture consisting of an N-terminal region containing functionally important cysteines and a C-terminal domain with up to 22 ankyrin repeats (ARs)[14,15]. The mature toxins are created from non-toxic precursors by posttranslational cleavage of a short N-terminal and a larger C-terminal domain by Furin-like proteases[16]. In solution, LTXs exist as monomers or dimers[17–19]. However, in the presence of calcium ($Ca^{2+}$) or magnesium ions ($Mg^{2+}$), they have been observed to spontaneously oligomerize and integrate into the membrane, forming tetrameric pores that allow the selective passage of cations[18]. The efficiency of this process is significantly increased in the presence of receptors[20]. Recently, we reported electron cryo-

[1]Institute for Medical Physics and Biophysics, University Münster, Münster, Germany. [2]Center for Soft Nanoscience (SoN), University Münster, Münster, Germany. [3]Institute of Physical Chemistry, University of Münster, Münster, Germany. [4]These authors contributed equally: B. U. Klink, A. Alavizargar. ✉e-mail: andheuer@uni-muenster.de; christos.gatsogiannis@uni-muenster.de

microscopy (cryo-EM) structures of both the α-LCT monomer and the δ-LIT dimer, shedding light on the overall domain organization of the LTX family[19]. However, the mechanism of LTX pore formation remained elusive, as we currently lack detailed structures of LTX tetramers, and pore events have only been visualized at low resolution[18]. Understanding the mechanism of LTX action holds significant medical relevance[21,22] and the potential to lead to the development of biotechnological applications and biopesticides[23].

Here, we show cryo-EM structures of the α-LTX tetramer in both the prepore and pore states. Combined with AlphaFold2 structural predictions and MD simulations, we elucidate the long-sought-after α-LTX pore and unveil a unique mechanism characterized by dramatic conformational changes during α-LTX transition into a cation-selective channel.

## Results
### Cryo-EM structures of α-LTX
We used cryo-EM to image α-LTX from the Mediterranean black widow spider (*Latrodectus tredecimguttatus*) in the presence of divalent cations to induce assembly into tetramers[18]. 2D classification revealed approximately 15% monomers, 19% dimers, 11% trimers, and 55% tetramers (Supplementary Fig. 1a). The tetramers exhibited a preferred orientation with their symmetry axis exclusively perpendicular to the air-water interface, thus providing only a single view of the protein. An initial 2D analysis revealed the presence of at least two distinct tetrameric states, both displaying a characteristic central channel in a wide and narrow conformation (Fig. 1a and Supplementary Fig. 1a; state 1 with 51 % and state 2 with 4% of particles, respectively). To address the preferred orientation issue, we collected multiple datasets with different tilt angles of the microscope stage from 0 to 60° and solved the structures of both states at 3.1 Å and 3.7 Å average resolution, respectively (Fig. 1a, b, Supplementary Figs. 1–4, Supplementary Table 1, and Supplementary Movies 1 and 2).

The architecture of each subunit of state 1 has a characteristic G-shape and is similar to that of the soluble α-LCT monomer and the δ-LIT dimer[19] (Fig. 1b, c and Supplementary Fig. 5). Each monomer is composed of an N-terminal four-helix connector domain (CD; residues E21–D115), a central helical bundle domain (HBD; residues S116–K350), a short β-sheet plug domain (PD; residues Q351–D453) and a C-terminal tail of 22 ARs (ARD; residues I454–G1195; Supplementary Fig. 5). Compared to the previous published soluble α-LCT monomer[19], neither large-scale conformational changes of individual subunits nor exposure of hydrophobic patches are required for tetramerization, indicating that state 1 corresponds to the prepore state of α-LTX. The four HBDs form the core of the tetramer with a central ~12–15 Å channel and the ARDs extend to the periphery, in a four-bladed windmill-like assembly (Fig. 1b). The tetramer in the prepore state is assembled by stronger interactions between the PD of one monomer and ARs 1–6 of the clockwise neighboring monomer and is further stabilized by weaker interactions between the HBDs and PDs of neighboring monomers (Supplementary Fig. 6b, d and Supplementary Table 4). We observe significant movements of the HBDs and ARDs relative to each other, inducing conformation heterogeneities and symmetry breaks in the potential four-fold symmetry (Supplementary Movie 3). Interestingly, in the prepore state, the CD is sandwiched in a pocket between the three other domains of the monomer and completely avoids interactions with neighboring monomers (Fig. 1c and Supplementary Fig. 5b).

In the second state, the HBDs are moved closer together, forming tight interactions that result in the narrow conformation of the central core (Supplementary Fig. 6c, f). The CD assumes a dramatically different conformation and is completely folded out of its pocket (Fig. 1b, c). The four N-terminal CDs are remodeled in an extended coiled-coil needle, whereas the C-terminal domains preserve their overall arrangement. This results in a characteristic mushroom-like

structure (Fig. 1c), similar to other pore-forming toxins[24–28], indicating that state 2 represents the long-sought pore-state of α-LTX. The distal end of the tetrameric coiled-coil needle is formed by the most N-terminal region of the protein (helices α1–α3 in state 1) and might correspond to the transmembrane region of the pore. This domain however is flexible and not resolved in the cryo-EM structure, as might be expected in the absence of a lipid environment or detergents.

### α-LTX inserts into membranes in an extended, mushroom-like conformation
It is well-established that α-LTX is able to integrate into biological membranes and this way forms cation-selective pores[20,29–32]. However, the underlying molecular mechanism and even the domains involved remained elusive. Despite significant efforts, we did not succeed in inducing efficient pore formation, even in the presence of different detergents or lipid nanodiscs. Detergents negatively affected complex integrity, whereas reconstitution in lipid nanodiscs was too inefficient for subsequent cryo-EM studies of the pore. Once we added α-LTX to liposomes, we noticed by negative stain EM a small number of successfully reconstituted α-LTX particles (Fig. 2a and Supplementary Fig. 7). This observation affirms α-LTX propensity for membrane insertion, but also aligns with previous studies suggesting that membrane insertion is highly inefficient in absence of receptors[20]. The particles show characteristic mushroom-like structures resembling the cryo-EM structure of state 2 with the (N-terminal) tip of their needle inserted into the membrane (Fig. 2a, b). We, therefore, conclude that the mushroom-like cryo-EM structure of state 2 indeed represents the pore state of α-LTX.

Using AlphaFold2, we obtained a prediction of the tetrameric assembly of N-terminal residues E21–E360 (consisting of the central CD and HBD, in the absence of ARDs). This model contains the structure of the unresolved tip of the needle (Fig. 2c and Supplementary Fig. 8), thereby complementing the cryo-EM structure (residues F146–G1195). In the overlapping region of the pore, the two structures agree very well (RSMD 0.9 Å). While attempts to predict the structure of the full-length α-LTX tetramer using AlphaFold2 were not successful, we obtained a complete model of the α-LTX pore by combining the cryo-EM structure (residues F146–G1195) with the AlphaFold2 prediction of the missing region including the CD (residues E21–R145) (Fig. 2d). These data allow a mechanistic description of the pore formation events.

During prepore to pore transition, helix α5 separates from the HBD and undergoes a reorientation of almost 180 degrees (Fig. 2e). This way, it forms an intriguing rigid tetrameric coiled-coil with the α5 helices of the other three monomers. In addition, helix α4 of the CD folds into a continuous helix with α5. In this manner, the coiled-coil elongates, positioning the N-terminal helices α1–α3 of the CD at its lower end to form the transmembrane domain (TMD) (Fig. 2e).

Direct comparison of the cryo-EM structure of the N-terminal helices α1–α3 in the prepore state with their AlphaFold2 prediction in the pore state suggests that these helices undergo dramatic conformational changes to assemble the TMD: In the prepore state, α1–α3 are isolated within each monomer, with a distance of ~60–80 Å to each other, and are bent into a total of five shorter sub helices (α1a, α1b, α2, α3a, and α3b; Fig. 2e and Supplementary Fig. 5). In the prediction of the TMD, they refold and arrange into a disulfide-stabilized cylindrical transmembrane bundle of 4 × 2 antiparallel helices formed by helices α1–α3 from all four monomers (Fig. 2e). This way, a hollow barrel is assembled.

The prediction of the distal end of the stalk is consistent with the expected features of a TMD of a pore-forming toxin: While the assembly presents a highly hydrophobic patch spanning about two-thirds of its outside surface, it creates a negatively charged inner cavity with an average diameter of about 15 Å (Fig. 2c). The domain fully spans the lipid bilayer, while the 4 × 2 helix bundle itself is only

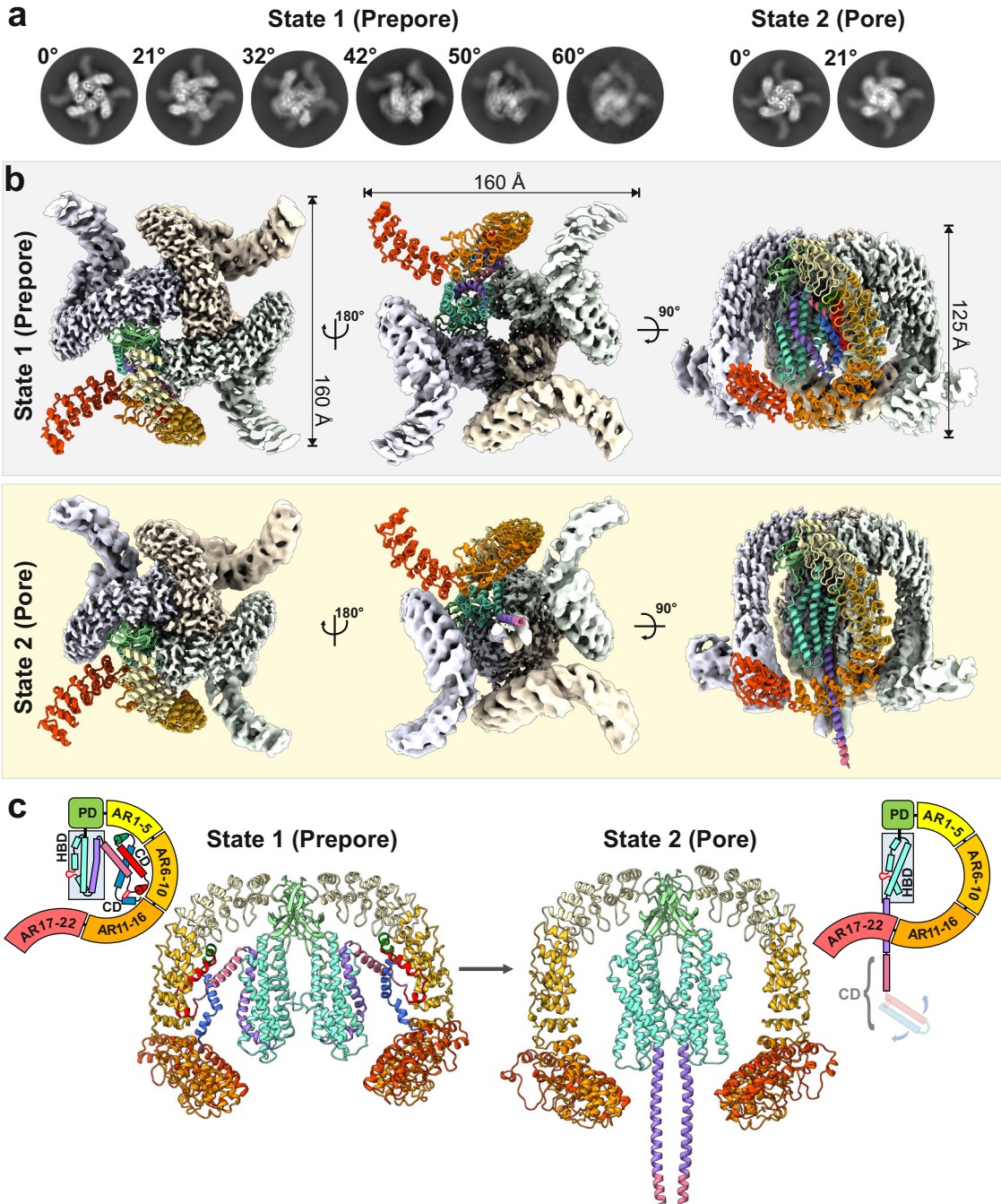

**Fig. 1 | Cryo-EM structure of α-LTX in two distinct tetrameric states. a** 2D class averages from datasets measured at different stage tilts. At low tilt angles, the presence of two distinct tetrameric states becomes apparent. Note the difference in diameter of the central channel. **b** Maps of the prepore (gray) and pore (yellow) state. For one monomer each, the derived molecular model is shown. **c** Molecular models of prepore and pore state of α-LTX. The side views depict two opposing subunits, colored according to their domain organization, as shown in the respective scheme. The first ~100 residues of the CD ("tip of the needle") are not resolved in the pore structure. CD connector domain, HBD helical bundle domain, ARD ankyrin-like repeat domain, PD plug domain.

penetrating with its hydrophobic patch. The outwards-facing third of the barrel is highly negatively charged, which might act as a stop signal, preventing further penetration of the stalk into the membrane (Fig. 2c). Such charged "stop-patches" have also been observed for other pore-forming toxins[26,33]. To further challenge the prediction, we performed MD simulations and the TMD remained stable in a POPC membrane supplemented with cholesterol and POPA (Supplementary Fig. 9 and Supplementary Movie 4). In the case of α-LTX, the partial integration of the 4 × 2 helical bundle leaves a ~5 Å opening at the interface between the 4 × 2 helix bundle and the tetrameric coiled-coil

(Fig. 2c) which is framed by the negatively charged residues E28, E32, E38, D93, and D97 (Supplementary Fig. 10a).

## The cation-selective α-LTX channel
Even after identifying the transmembrane region within the CD, the translocation path of cations was at first glance not obvious. Interestingly, the HBDs, the tetrameric coiled-coil, and the TMD form a ~175 Å continuous channel (Fig. 3a). Moreover, the HBDs provide a loop near the coiled-coil entrance, each displaying two aspartate residues (D209 and D210), which create a negatively charged pocket (Fig. 3b). It was

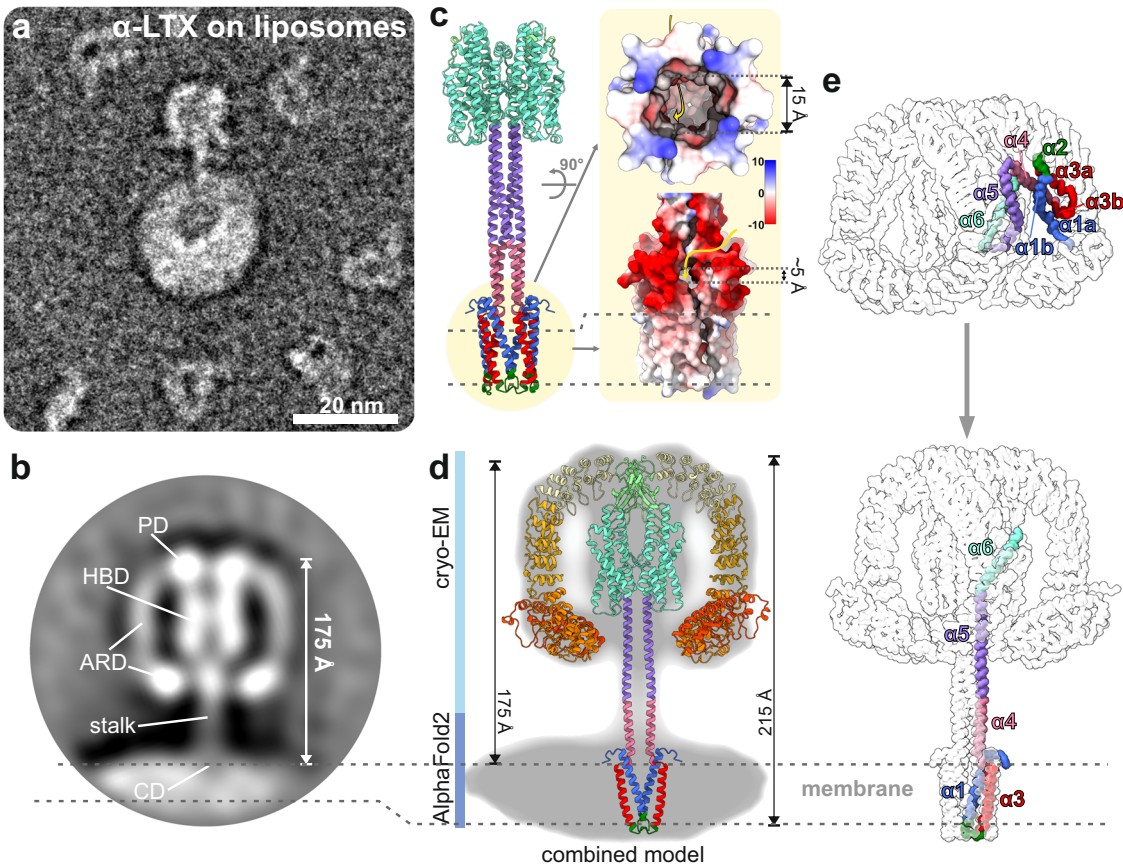

**Fig. 2 | Molecular architecture of the α-LTX pore. a** Representative α-LTX particle on a POPC-liposome in negative stain EM. From 988 micrographs, a total of 108 similar particles were used to create a 2D class average as shown in (**b**). The individual domains of the toxins are highlighted (CD connector domain, HBD helical bundle domain, PD β-sheet plug domain, ARD ankyrin repeat domain). **c** AlphaFold2 prediction of a tetrameric assembly of residues E21–E360 of α-LTX. The surface coulombic electrostatics [kcal/(mol·*e*)] indicates a membrane-spanning domain at its N-terminus. **d** Molecular model of the complete α-LTX pore obtained by combining the cryo-EM structure and the AlphaFold2 prediction overlaid in the negative stain average of α-LTX pore on liposomes. **e** Structural rearrangements of the CD lead to the formation of an extended coiled-coil and the TMD.

tempting to speculate that $Ca^{2+}$ might enter the continuous channel through this site, in analogy to a "cation selectivity filter", and subsequently pass through the coiled-coil towards the TMD.

Interestingly, the pore-forming toxins Vip3 from *Bacillus thuringiensis* share some overall similarities with α-LTX[24,27]. Vip3Aa undergoes dramatic conformational changes upon proteolytic cleavage, forming a tetrameric N-terminal coiled-coil stalk with its flexible end penetrating the membrane, thereby creating a cation-selective channel. However, α-LTX and Vip3Aa have no additional structural similarities (Supplementary Fig. 11). Vip3Aa includes a cation binding site close to the proximal entrance of the coiled-coil but according to recent studies, the stalk itself does not serve ion transportation. The first N-terminal α-helix of Vip3Aa at the lower end of the stalk is crucial for both insecticidal activity and liposome permeability and could form an ion-permeable channel in contact with the membrane[24,34,35].

In α-LTX, the inner radius of the tetrameric coiled-coil is consistently less than 2 Å (Fig. 3a), which is smaller than a $Ca^{2+}$-oxygen binding distance and therefore too narrow for $Ca^{2+}$ to pass through. Furthermore, MD simulations suggest that the HBD pocket above the coiled-coil stalk in α-LTX strongly binds $Ca^{2+}$, but the affinity was so strong that even with an applied electric potential difference of 200 mV, a trapped cation was never released throughout the time frame of the simulation (Fig. 3b). Instead when a pulling force was applied to push the bound $Ca^{2+}$ ion into the stalk, the cation quickly exited sideways (Supplementary Fig. 10b). Similarly, no $Na^+$ was observed to enter the stalk (Fig. 3b). Thus, similar to Vip3Aa, it is highly

unlikely that the stalk can transport ions, as further supported by its enormous stability, precluding any significant conformational changes involving widening of its inner diameter and passage of ions (Supplementary Fig. 12).

Since the structural properties of the stalk do not allow for ion transport, we ask whether the predicted model of the TMD region, whose stability was confirmed by MD simulations, could form a cation-selective channel. Here, we use again MD simulations of the TMD region together with the distal end of the stalk and analyze this hypothesis by analyzing its behavior in the presence of different ions.

$Na^+$ and $Ca^{2+}$ ions were able to pass in large amounts; representative example trajectories are shown in Fig. 3c. The higher charge of $Ca^{2+}$ compared to $Na^+$ has two different implications. First, for $Ca^{2+}$ the resulting stronger driving force after applying the electric field yields a stronger directional bias and thus higher selectivity. Second, $Ca^{2+}$ has more localized positions than $Na^+$ in the simulation, resulting in fewer and longer-lasting membrane crossing events (Fig. 3c, d, Supplementary Tables 2 and 3, Supplementary Fig. 13, and Supplementary Movie 5). These results are fully consistent with the reported properties of cation-selective LTX pores in vivo[31]. Cartoon representations of the simulation of the TMD (Fig. 3d) clearly show the integrated positions of the cations during transport. By studying the height-dependent 2D-resolved $Ca^{2+}$ density, it is even possible to identify the four entrance points and subsequent channels (Fig. 3d). The entrance gate is localized at the 5 Å opening at the interface between the stalk and the TMD (Fig. 2c and Supplementary Fig. 10a),

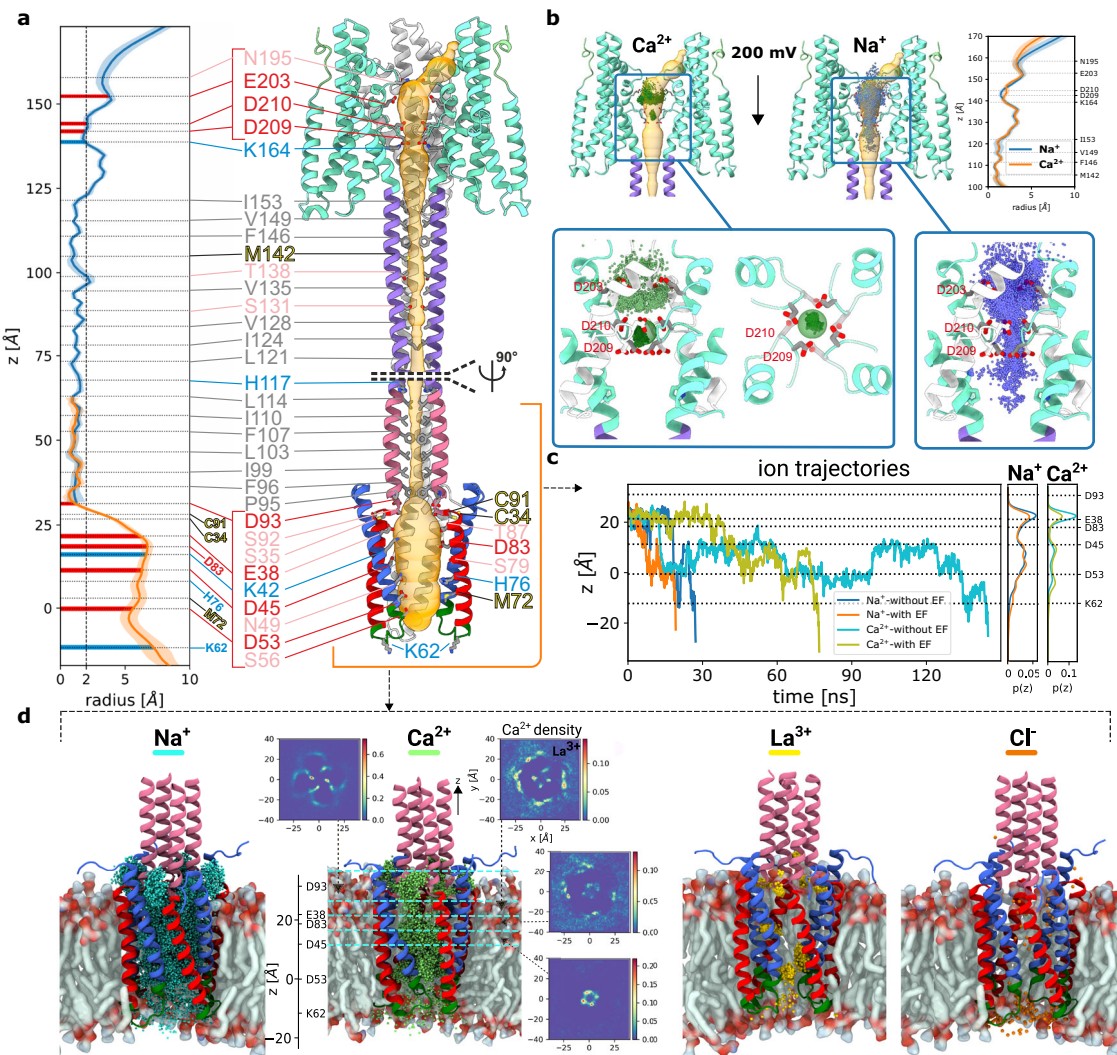

**Fig. 3 | MD simulations of the pore structure of α-LTX. a** Cartoon representation of the pore structure of α-LTX (residues E21–E360). The pore is represented as an orange surface. Residues facing the pore volume are represented as sticks with labels colored according to their biochemical properties. The pore profiles from the two simulations (residues C91–Y260 for the upper part and the stalk; residues E21–S116 for the membrane protein part) are shown as blue and orange lines, respectively. **b** Cartoon representation of the Na⁺ and Ca²⁺ positions in the HBD, respectively, obtained from a superposition of many simulation snapshots with an applied electric field. Also, the resulting pore radius is shown together with its standard deviation. **c** Representative ion trajectories, correspond to typical permeation events of the TMD by Na⁺ and Ca²⁺ ions, with and without an applied electric field. Additionally, shown are ion density profiles obtained by averaging all successful permeation events. **d** Cartoon representations of simulations of the membrane part with different ions (Na⁺, Ca²⁺, La³⁺, and Cl⁻) with a superposition analogous to **b**. The Ca²⁺ ionic densities in different slabs in the *xy*-plane are shown (5 Å and 10 Å widths along *z*, respectively).

which indeed seems to fulfill important functions as a cation gate. La³⁺ ions are trapped at the entrance hole towards the cavity and block the pore (Fig. 3d), in agreement with previous in vivo observations[15,31,36,37]. For Cl⁻, the strong negative charge of the entry region (Fig. 2c and Supplementary Fig. 10a) is prohibitive.

Taken together, our findings contribute to a robust model that aligns with the extensive literature on the electrophysiological properties of the cation-selective α-LTX channel, including its mechanisms of ion transport and channel gating.

### Formation of tetramers prior to membrane insertion

α-LTX purified in the presence of Ca²⁺ consists of a mixture of all oligomeric prepore states from monomer to tetramer and a small population of the tetrameric pore assembly (Supplementary Fig. 1a). The existence of trimeric prepore species suggests that the formation of a tetrameric prepore from two dimers may not be the sole route of complex assembly, as previously proposed[38]. A stepwise assembly of the tetrameric prepore can easily be rationalized by the involved

subunit interactions, which are almost exclusively formed between the PD of one monomer with the AR domain of the clockwise neighboring monomer (Supplementary Fig. 6).

Tetramerization was blocked in the N4C-mutation of α-LTX by insertion of residues VPRG before the first AR[15]. This mutant abolishes its ability to form pores and is therefore extensively used in the field to selectively study the receptor-mediated actions of α-LTX[17,38–40]. The tetrameric prepore is well resolved in this region and shows that the introduced VPRG residues are placed N-terminal of the last β-sheet (β7) of the PD domain, disturbing the crucial intermolecular oligomerization site between PD and neighboring ARD (Supplementary Fig. 14). This explains why this mutant loses its ability to oligomerize and to form ionophores[17,37], while it retains the ability to bind and activate receptors[17,40].

Consistent with previous functional studies[41], our data support the notion that the formation of the tetrameric prepore assembly is a prerequisite for the formation of the pore state (Supplementary Fig. 6a). The dramatic prepore to pore rearrangement relies mostly on

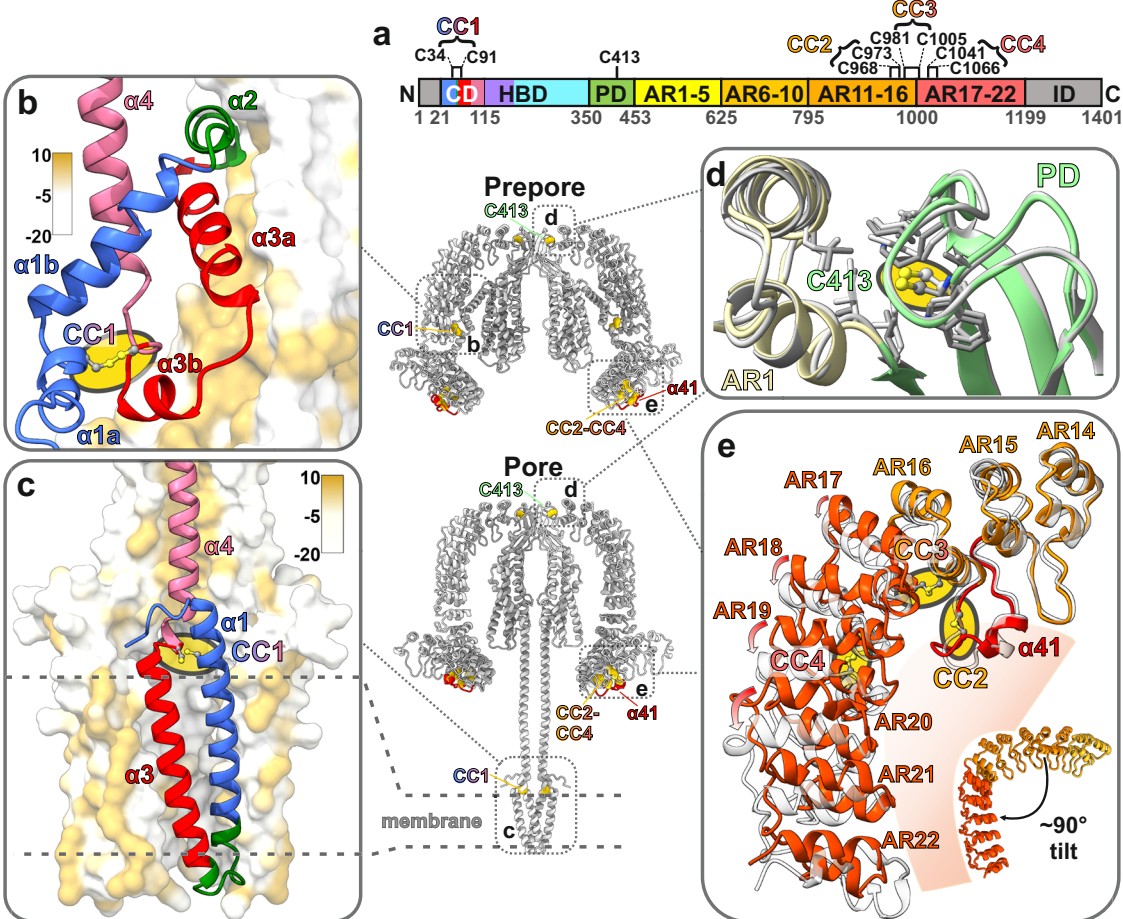

**Fig. 4 | α-LTX stabilization by cysteines and disulfide bonds. a** Position of cysteines in α-LTX. Disulfide pairs (CC1–CC4) are marked with braces. **b** CD in the prepore state. The interacting AR domain is contoured by its surface hydrophobicity. **c** CD in the pore state. The interacting CDs from different α-LTX monomers are contoured by surface hydrophobicity. **d**, **e** Cys413 and disulfide bonds in the AR domain. The prepore state is colored and the pore state in light gray.

the successful formation of the rigid tetrameric coiled-coil by helices α4 and α5. The stability of the coiled-coil is quantified by studying the increase in free energy when slowly removing one monomer of the prepore structure as reflected by an increasing RMSD as compared to the tetrameric coiled-coil structure in the pore state (Supplementary Fig. 12). Moreover, the interactions within the tetrameric coiled-coil in the pore state are about an order of magnitude stronger than the PD/ARD interactions, which have the strongest contribution in the prepore state (-30 kcal/mol vs ~3.3 kcal/mol, respectively) (Supplementary Table 4).

Importantly, our structures suggest that only towards the formation of the pore state, the HBDs move closer together to form several stabilizing contacts (Fig. 1b and Supplementary Fig. 6c, f), including the prominent $Ca^{2+}$ binding site directly above the coiled-coil stalk (Fig. 3b). This site binds $Ca^{2+}$ with such high affinity in our MD simulations that it cannot easily be released (Fig. 3b and Supplementary Fig. 10b), suggesting that $Ca^{2+}$ binding supports the dramatic conformational rearrangement of the prepore tetramer into the pore state. Furthermore, the observation that the cavity radius in the upper part of the HBD pocket is smaller for $Ca^{2+}$ than for $Na^+$ (Fig. 3b) may suggest a $Ca^{2+}$ stabilizing mechanism for the pore state.

**Role of disulfide bridges for the transition to the pore state and receptor binding**

Previous biochemical studies suggest that disulfide bond reduction abolishes the activity of α-LTX and its binding to receptors[11,15]. α-LTX

contains a total of 9 cysteine residues, two of which are in the N-terminal CD, one in the PD, and six in the C-terminal AR15-19 (Fig. 4a).

The N-terminal conserved cysteines of the CD were mutated to serine (C34S, C91S, and C413S) and were found to be essential for α-LTX action[15]. Our data confirm that the conserved C34 and C91 are involved in a stabilizing disulfide bridge (CC1) between helices α1 and α3 of the CD, both in the prepore and the pore states (CC1; Fig. 4b, c). It should be emphasized that these helices form the membrane-spanning α-LTX channel (Fig. 2e). Although the fold of helices α1–α3 is remarkably different between the two states (Fig. 4b, c), CC1 plays an important role as a stabilizing connection for this assembly and thus for the prepore to pore transition, as it stays intact during the refolding and massive reorientations of up to ~190 Å from the prepore to the pore state. Alphafold2 predictions suggest that the stabilization of the membrane-spanning domain by CC1 is conserved throughout the LTX family (Supplementary Fig. 15a).

In contrast, C413 does not form a disulfide bond but is a central part of a hydrophobic pocket connecting PD and ARD in α-LTX (Fig. 4d). Loss of function upon mutation of C413 to a less hydrophobic serine is thus not due to lack of disulfide potential, but rather due to an altered protein fold.

Furthermore, our structures reveal that the six C-terminal cysteines in AR15–AR19 of α-LTX can form three disulfide bonds (CC2, CC3, and CC4), which might be important for receptor binding and toxin specificity (Fig. 4e and Supplementary Fig. 15), since they are

located in the region that contains the interaction site for the receptor Latrophilin[42,43]. Removal of ARs 15–22 containing CC2, CC3, and CC4 results in a complete loss of α-LTX stimulating activity[44].

CC2 is the most N-terminal disulfide bond in the ARD (C968–C973) in α-LTX and is located within an extended loop between ARs 15 and 16, succeeding the short helix α41 located within this loop (Fig. 4e). This extended loop between AR 15 and 16, in combination with an unusually short loop between AR16 and AR17, introduces a rotation in the alignment of ARs to each other by almost 90°. CC3 and CC4 in α-LTX are located directly after the extended loop between ARs 15 and 16, which qualifies them to stabilize this characteristic bending point in the middle of the ARD tail (Fig. 4e).

Interestingly, the position of the characteristic loop and CC2 in the AR tail are conserved in the vertebrate-specific α-LTXs, but have a different arrangement in the crustacean-specific α-LCT and the insect-specific α-LIT. These do not contain such an extended loop between AR 15 and 16, but a loop is inserted within AR18. This loop is also stabilized by a disulfide bond, which we call CC2* in α-LCT and α-LIT for better clarity. AlphaFold2 predictions suggest that this distinct positioning of the disulfide-stabilized loop also results in a different bending of the AR tail (Supplementary Fig. 15b). The insect-specific δ-LIT contains only 13 ARs, i.e., its tail is truncated and does not contain these disulfide bonds[19], but instead a highly negatively charged domain that could represent an alternative receptor binding motif (Supplementary Fig. 15b).

The ARDs containing the receptor binding sites are otherwise conserved across the members of the LTX family, known for being highly phylum-specific. It is conceivable that the consistent positioning of disulfide bonds CC3–4, coupled with variable positions of the loop containing CC2 or CC2*, plays a crucial role in receptor binding. The architecture of the loop, along with the curvature and length of the ARD tail, emerges as the most likely set of factors determining LTX target specificity.

## Discussion

Our study provides mechanistic insights into the process through which α-LTX integrates into presynaptic membranes and creates a selective cation-permeable channel that facilitates calcium influx, triggering uncontrolled neurotransmitter release. The results presented here explain numerous long-standing questions regarding the mechanism of LTX action accumulated during the last decades. Based on our findings and previous work in the field, we propose a model for α-LTX action at the presynaptic membrane (Fig. 5).

An α-LTX precursor is cleaved by furin-like proteases at its N- and C-termini, producing an activated soluble G-shaped toxin monomer[16]. Four monomers assemble stepwise via interactions between the PD and ARD domains to form the tetrameric prepore state that is competent to initiate membrane insertion. This process is most probably supported by the binding of receptors to the ARD-tails, which might be crucial to bringing the complex in proper orientation and distance to the membrane. Noteworthy, the formation of pores is possible in pure lipid membranes[20,32] (Fig. 2a), but the presence of receptors significantly increases the occurrence of pores[20].

The prepore displays a high degree of conformational flexibility in the curvature and/or relative orientation of different subunits and domains, resulting in "breathing motions" of the central channel (Supplementary Movie 6). We believe that narrowing of the channel, as seen in subsets of our cryo-EM data, marks the first prerequisite step to initiate pore formation: only within a sufficiently narrow channel, the four loop regions containing aspartates D209 and D210 are close enough to form a central high-affinity cation binding pocket as seen in the pore state (Fig. 3b and Supplementary Movie 6). This process, which most probably progresses in a stepwise manner, also pulls the termini of helices α6 close enough together to initiate the formation of

the tetrameric stalk. It is conceivable that the stabilization of the narrow conformation by Ca²⁺ ions in the D209/D210 binding pocket is the determining factor for the described Ca²⁺-dependence of α-LTX pore formation[36,41]. Moreover, in the prepore state, the helices of the N-terminal CD are tightly packed and protected within a groove formed between the HBD and the ARD-tail, with helices α1−α3 being fragmented, but stabilized by a conserved disulfide bond. Ca²⁺-dependent narrowing of the central HBD channel, combined with the tetrameric stalk formation starting at the hinge between helices α6 and α5, might also destabilize the groove harboring the CD.

Upon opening of the groove, the assembly of the tetrameric stalk can now proceed by spontaneous rearrangements of the CDs elongating the prearranged N-termini of the tetrameric coiled-coil stalk, up to its final length of about 15 nm. This process might resemble the assembly of SNARE complexes which promote vesicle fusion by the formation of a highly stable tetrameric coiled-coil[45]. The four N-terminal α-helical pairs of the CD, each stabilized by the conserved disulfide bond that remains intact during this conversion, come close to each other at the low end of the stalk. Together the CDs insert into the membrane to assemble the transmembrane channel. The prepore-to-pore transition converts the flat windmill-like prepore complex into an extended mushroom-like pore conformation (Fig. 5 and Supplementary Movies 7 and 8).

Notably, the HBD, the stalk, and the TMD thereby form a continuous channel. The stalk is however not permeable for ions, but might rather have functions as a molecular ruler: α-LTX receptors exhibit an elongated shape, with the extracellular domain protruding far from the membrane[46,47]. Upon receptor binding, the coiled-coil stalk possesses an appropriate length to span this distance and deliver the CD to the presynaptic membrane (Supplementary Movie 7). In light of the substantial height of the pore state of approximately 175 Å protruding from the membrane, these dimensions remain in accord with the ~20 nm width of the synaptic cleft[48].

We identified the position of a side-entry gate that is selective for monovalent and divalent cations and located at the interface between the coiled-coil stalk and the transmembrane CD. Ca²⁺ ions can enter this gate from the side of the stalk directly above the upper leaflet of the presynaptic membrane and pass through the transmembrane pore (Supplementary Movie 7). This mechanism mimics Ca²⁺ influx via voltage-gated Ca²⁺ channels during an action potential and triggers massive synaptic vesicle exocytosis. High-resolution structures of the transmembrane region and variants in a near-to-native environment will be of significant importance to confirm these results, provide more detailed insights into α-LTX ion translocation, and shed light on toxin-lipid interactions.

Our study reveals how the soluble α-LTX tetramer undergoes a metamorphosis to form a cation-permeable transmembrane pore and allows us to understand key steps of α-LTX action at the molecular level. Important outstanding questions include how binding to receptors increases the efficiency of toxin pore formation and the receptor-dependent mechanisms that determine phylum-specificity among different members of the latrotoxin family.

## Methods

### Protein purification

The α-LTX sample used in the study was purchased from Alomone Labs (Cat. LSP-130). It was isolated from *L. tredecimguttatus* (black widow spider) venom by modifying a previously reported protocol[49,50]. The lyophilized protein was dissolved in sample buffer containing 25 mM Tris-HCl pH 8.0, 150 mM NaCl, 0.1 mM CaCl₂, and 0.2 mM MgCl₂ and further purified by size exclusion chromatography on a Superdex 200 increase 5/150 column (Cytiva) equilibrated in sample buffer. The eluates were fractionated and confirmed by SDS-PAGE and negative stain electron microscopy (EM).

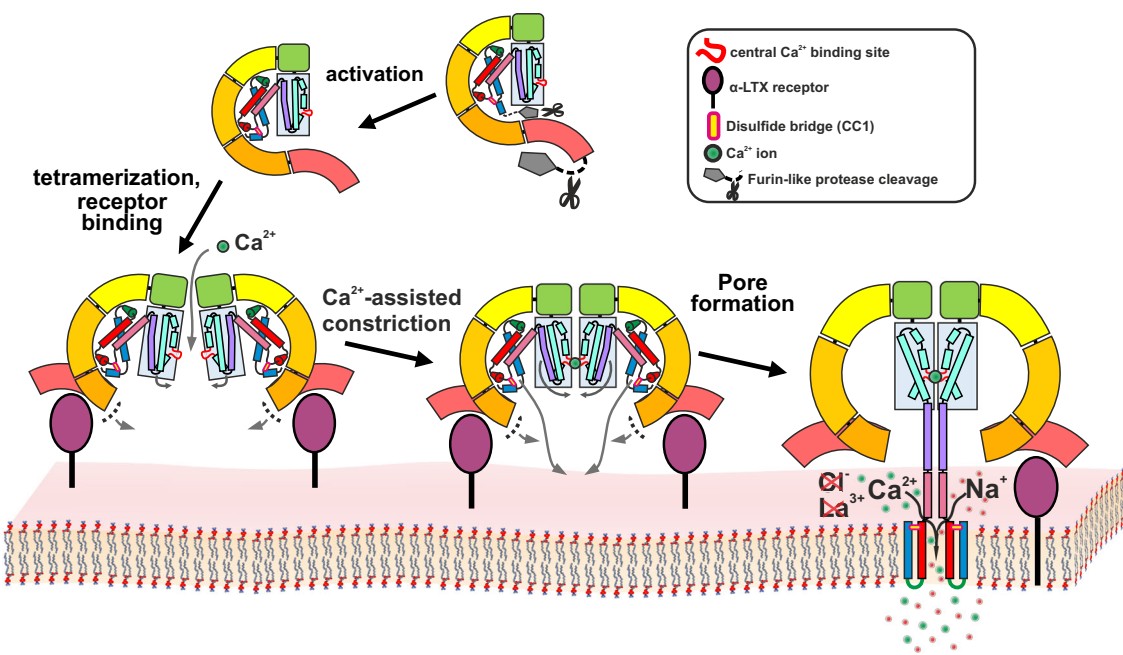

**Fig. 5 | Schematic model of α-LTX activation, tetrameric prepore assembly, and transition to a cation permeable channel.** The precursor of α-LTX is activated by proteolytic cleavage of its inhibitory terminal domains in the venom gland. Tetramerization, a prerequisite for pore formation, may be enhanced through association with presynaptic receptors. A central Ca²⁺ binding site brings the HBD domains into close proximity. The C-terminal helices of the HBDs, along with the CD domains, reorient to form a tetrameric coiled-coil stalk. The N-terminal helices of this stalk, stabilized by disulfide bridges, insert into the membrane, forming a cation-permeable transmembrane channel.

## Liposome preparation and binding assay

For liposome preparation, 10 mg 1-palmitoyl-2-oleoyl-sn-glycero-3-phosphocholine (POPC) lipids (Avanti Polar Lipids) were dissolved in a 1:1 methanol/chloroform mixture and lyophilized as a thin lipid film. The film was then rehydrated with 1 ml buffer-detergent mix, containing 1% Triton-X 100 in sample buffer. After dissolving, the mixture was incubated in a glass vial with 10% (w/v) Bio-Rad Bio-Beads SM-2 Adsorbent Media (Cat No. 152-3920) for 30 min at 4 degrees. Subsequently, Bio-Beads were added to a total of 30% (w/v), and the mixture was incubated overnight at 4 °C to remove all traces of detergent. As an alternative, we also prepared liposomes by the extrusion method, without using detergents: a suspension of 10 mg/ml POPC was homogenized by passing 21 times through a polycarbonate membrane with a 0.2 µm pore size in a mini extruder (Avanti Polar Lipids).

For the binding assay, the supernatant was diluted with sample buffer complemented with 5 mM CaCl₂ and 0.5 M KCl and incubated with α-LTX at final concentrations of 0.1 mg/ml liposomes and 0.02 mg/ml α-LTX for four hours at 310 K. The membrane incorporation of α-LTX was then analyzed by negative stain EM.

## Negative stain EM analysis

For negative stain EM sample preparation, 4 µl of the protein sample, diluted to a concentration of 0.005–0.02 mg/ml, was applied to a glow-discharged carbon-coated copper grid, and incubated for 2 min at room temperature. The excess protein solution was removed by blotting with Whatman (grade 4–5) filter paper, followed by washing with 2 × 10 µl deionized water or sample buffer, and 2 × 10 µl 0.75% uranylformate solution. The final uranylformate droplet was incubated for 45–60 s before blotting and air drying of the grid.

Image acquisition was performed using a Talos L120C G2 TEM operating at an acceleration voltage of 120 kV. Datasets were acquired with a 4k × 4k CETA-F scintillator camera at a defocus of −1 µm and a pixel size of 1.2 Å/px.

## Sample vitrification and cryo-EM data acquisition

For preparing the grids for cryo-EM, 4 µl of α-LTX sample at a concentration of 0.5 mg/ml were applied onto a freshly glow-discharged UltrAuFoil R2/1 holey gold grid (Quantifoil). After the removal of excess liquid, the sample was vitrified in liquid ethane using a Vitrobot II automatic plunge-freezer (Thermo Fisher Scientific).

Datasets were acquired with a 300 kV Titan Krios G4 microscope (Thermo Fisher Scientific) equipped with an E-CFEG, a Selectris X energy filter, and a Falcon 4i direct electron detector operated by the software EPU (Thermo Fisher Scientific). A total of ~90.000 micrographs in ten sub-datasets at different stage tilt angles from 0° to 60° were collected in electron event representation mode (EER) at a nominal magnification of 215k, corresponding to a pixel size of 0.58 Å/px. The majority of micrographs were collected at a defocus of −0.3 to −1.7 µm. The Selectris X energy filter was used for zero-loss filtration with an energy width of 10 eV. A total dose of ~50 e⁻/Å² (estimated shortly after freshly flashing the cold-FEG) was aimed for by adjusting the exposure time of each sub-dataset to an appropriate value (between 3 s and 4 s per micrograph). The details of dataset collection are summarized in Supplementary Table 1. Micrographs in which no particles were picked during image processing (see below) are not shown in the statistics.

## Image processing and 3D reconstruction

EER movies were motion-corrected in RELION4[51] using its own Motioncorr2[52]-like algorithm. In the first step, 2,436,143 particles were picked from the motion-corrected micrographs by crYOLO[53] based on a generalized picking model and extracted with a final window size of 560 × 560 pixels. CTF estimation was performed using CTFFIND4[54]. Based on the estimated defocus values and resolution limits, 75,968 good micrographs were selected for further processing.

An initial 2D classification was performed in RELION4 at a pixel size of 2.32 Å/px. Classes displaying incomplete tetramers, and contaminants were removed, and the remaining 974,801 particles were used to generate an initial 3D model. From a range of four different tilt

angles, 1000 micrographs (for each tilt-angle) with the most particles after 2D classification were used to manually train crYOLO for an optimized picking model. Another 2,694,664 particles were picked with this model from the datasets with 21°–60° tilt and also classified in 2D. After combining the best particles with the first set of 2D classified particles and removal of duplicates, 1,162,294 particles were reextracted at 1.69 Å/px. A 3D Refinement in RELION4 reached 3.73 Å resolution. In a subsequent 3D classification into 7 classes, one class clearly showed a largely different (Pore) conformation than the other (Prepore) classes. We used this pore class with 70,971 particles and the three best prepore classes with a combined amount of 746,650 particles for further processing of the pore and the prepore state, respectively. Both for pore and prepore conformations, initial 3D refinements were performed with global angular sampling, and in later stages of processing, local refinements with C1 symmetry and symmetry relaxation to C4, as implemented in Relion4, were used to resolve potential misalignments of pseudo-symmetric particles. For generating Supplementary Movies 3 and Movie 6, particles from all fully assembled prepore classes were used to represent a more complete conformational landscape of prepore motions (classes 1, 3, 5, 6, and 7 with a total of 968,962 particles; Supplementary Fig. 2).

The pore state was further processed at a full pixel size of 0.58 Å/px. The first 3D refinement reached 4.16 Å resolution, and the 3D map was improved to 3.69 Å resolution by one round of CTF and aberration refinement, Bayesian polishing, and another round of CTF and aberration refinement, each followed by 3D auto-refine in RELION4. The obtained map had a strong resolution gradient and presented strong heterogeneity, particularly in the C-terminal part of the ARD regions. This was accounted for by a final multi-body refinement with one body for each C-terminal half of the ARDs and one body for the central part of the complex, which improved the resolution in the center to ~3.2 Å. To maximize map interpretability, we also generated sharpened maps in RELION4 and density-modified maps in Phenix[55]. Since each map provided the best interpretability in a different region of the structure, we combined a total of nine maps with Phenix.combine_focused_maps. This approach analyzes the map-model correlation of multiple maps to identify and superpose the best parts of each map to create a composite map[55]. We used this map together with the individual maps for model building and map interpretation (Supplementary Fig. 2).

The prepore state was processed using a similar approach, but at a pixel size of 1.16 Å/px, and with the difference that after a first round of CTF and aberration refinement, a second 3D classification into four classes was used to further remove conformational heterogeneity. A class containing 442,105 particles displaying high-resolution features up to 3.78 Å was selected for subsequent Bayesian polishing and another round of CTF and aberration refinement, each followed by 3D auto-refine in RELION 4. This improved the resolution to 3.12 Å. A final multi-body refinement with one body for each C-terminal half of the ARDs and one body for each N-terminal part improved the core resolution to ~2.71 Å, and the C-termini also became more interpretable. As for the pore state, we generated a combined focused map using Phenix, here with a total of 20 individual maps to generate an optimized composite map.

## AlphaFold2 structure prediction

A prediction of the monomeric mature α-LTX (UniProtKB ID: P23631 (LATA_LATTR)) was performed using AlphaFold2[56]. It resembled our previously solved structures of α-LCT and δ-LIT (PDBIDs: 7PTX and 7PTY, respectively), with the same domain architecture and a similar G-shaped arrangement[19]. The N-terminal CD domain had a low confidence in the prediction, and assumed a conformation that is not consistent with our cryo-EM prepore structure. Attempts to predict the full tetrameric assembly using the "multimer" mode of AlphaFold2 were not successful, but a prediction of residues E21–E360, corresponding to the CD and HBD, assumed a conformation in which the CD

domain and helix α5 of the HBD form an elongated tetrameric coiled-coil stalk. This conformation largely differs from the prediction of the monomer, but instead is consistent with the pore state as identified by cryo-EM and negative stain EM. For all AlphaFold2 predictions, five models were generated and the model with the highest prediction score was used for further analysis.

## Model building and validation

The AlphaFold2 prediction of monomeric α-LTX was used as an initial model for building the cryo-EM prepore state. Initially, the model was copied to the four subunits of the experimental tetramer and relaxed into the density by fragmented rigid-body fitting and subsequent real-space refinement in Phenix[55,57]. For the regions where the predicted model did not fit well (particularly the CD), the model was rebuilt from scratch using the model editing software COOT[58,59]. The resulting model was further refined using a combination of COOT and Phenix. The model quality was evaluated by the validation tools implemented in Phenix[60] and the wwPDB validation server[61]. Multiple rounds of the above adjustments were performed until the model sufficiently described the experimental map.

An initial model of the pore state was created by combining residues N105–K350 from the AlphaFold2 prediction of the tetrameric N-terminus with the missing C-terminal part from the monomer prediction. As for the prepore state, fragmented rigid-body fitting and real-space refinement in Phenix followed by multiple rounds of model refinement with COOT and Phenix were used to create the pore model.

## Visualization and analysis of cryoEM maps and models

Visualization, analysis, and figure preparation were done with ChimeraX (UCSF)[62,63] and CorelDRAW (https://www.coreldraw.com). Local resolution gradients within a map were calculated with RELION4 and visualized with ChimeraX. 3D angular distribution plots were generated in Relion4. 2D histograms of the angular distribution were generated using angdist[64]. The 3D Fourier shell correlation of cryo-EM maps was calculated using the remote 3DFSC processing server[65]. Interfaces within α-LTX tetramers and their solvation-free energies were analyzed by the Pisa server[66].

## MD simulations: system preparation

Starting from the AlphaFold2 prediction, we modeled, first, the tetrameric coiled-coil domain, namely the stalk (residues C91–D155) and, second, a part of the HBD region and the tetrameric coiled-coil region (residues C91–Y260). The proteins were then solvated with water and ions using the CHARMM-GUI solvation builder[67]. For the first case, 0.15 M NaCl was used, in the second case an additional simulation with 0.15 M CaCl$_2$ was performed. A similar procedure was followed to model the cryo-EM structure of the pore state (residues N105–Y260) solvated with water and 0.15 M NaCl.

To model the membrane protein part, the residues E21–S116 were inserted in a 7:2:1 POPC:CHOL:POPS lipid mixture, corresponding to a synaptic membrane, which was then solvated with water and ions, using CHARMM-GUI membrane builder[67]. Three systems were simulated with three different ion concentrations of 0.15 M NaCl, CaCl$_2$, and LaCl$_3$. Default protonation states of the titratable residues were used for the modelled proteins.

## MD simulations

All MD simulations were performed using the 2019.6 version of GROMACS[68,69]. The TIP3P water model was used to describe the water molecules[70]. Periodic boundary conditions were applied in all directions. The system sizes were large enough to avoid protein-protein interactions via periodic images, involving system sizes between approx. 90,000 and 200,000 atoms. Long-range electrostatic interactions were treated with the use of the particle mesh Ewald method[71], with a cutoff distance of 1.2 nm and a

compressibility value of $4.5 \times 10^{-5}$. The van der Waals interactions were treated using cut-off schemes with a cutoff distance of 1.2 nm, which are smoothly truncated between 1.0 nm and 1.2 nm. The constant pressure was maintained at 1 bar by coupling the system to the Berendsen barostat[72] in equilibration and Parrinello–Rahman[73] barostat in production simulations, using the semi-isotropic pressure scheme for the membrane-protein system. The temperature was controlled at 310 K for all systems otherwise stated by coupling the system to the Nosé-Hoover thermostat[74]. The LINCS algorithm was utilized to constrain the bonds[75]. All systems were first minimized and subsequently equilibrated using initially the NVT (500 ps) and then the NPT (16 ns) protocol in multiple steps. During the course of equilibration, restraints ($1000\,kJ\,mol^{-1}\,nm^{-2}$) were applied to the protein so that the membrane and solvent were equilibrated around the protein. The production simulations were performed for 1200 ns using a time step of 2 fs.

For the tetrameric coiled-coil domain (residues C91–D155), we additionally performed a simulation starting from the equilibrated structure (50 ns) at 310 K and heated up the system gradually to reach the temperature of 400 K and continued the simulation for 240 ns. The structure at the end of the simulation was used as the starting structure for the simulation with a temperature of 310 K, which was done for 680 ns.

For the HBD region together with the tetrameric coiled-coil, as well as for the TMD inside the membrane, we additionally performed simulations under the electric field corresponding to the electric potential difference of 200 mV and 100 mV across the simulation cell, respectively. These simulations were performed for 1 μs and for systems with $Na^+$ and $Ca^{2+}$ ions.

### Steered MD simulations

To check the stability of the tetrameric coiled-coil (residues C91–D155), we applied harmonic force with a force constant of $1500\,kJ\,mol^{-1}\,nm^{-2}$ to the center of mass of the opposite chains to increase the distance between them. The rate was chosen as 0.01 nm per ns. One structure was then selected in which the chains have been opened so that the alpha helices in the chains have not been distorted and then this structure was simulated for 1 μs.

To pull the $Ca^{2+}$ ion inside the HBD and subsequently into the tetrameric coiled-coil, a harmonic pulling force was applied to the ion with a force constant of $1500\,kJ\,mol^{-1}\,nm^{-2}$ and the rate of 0.1 nm per ns opposite to the direction of the z-axis.

### Metadynamics simulations

The metadynamics simulation was performed on the combination of the cryo-EM structure of the prepore state, using residues C91–D155 (one chain) and the corresponding coiled-coil region of the pore-state (three chains) predicted by AlphaFold2, employing version 2019.6 of GROMACS[68,69] and version 2.6.4 of Plumed[76]. The collective variable was chosen as the RMSD of the chain, taken from the cryo-EM structure, with respect to the corresponding chain, predicted from AlphaFold2. During the simulation, several restraints were applied: (1) upper wall potential with the force constant of $1000\,kJ\,mol^{-1}\,nm^{-2}$ to the RMSD with the maximum value of 2.1 Å. (2) Lower wall potential with the force constant of $1000\,kJ\,mol^{-1}\,nm^{-2}$ to the number of alpha helices. This number was constrained to be larger or equal to 235. (3) Lower wall potential with a force constant of $1000\,kJ\,mol^{-1}\,nm^{-2}$ to the distance between the beginning and end residue of the cryo-EM part of the structure with the lowest value of 3.8 nm. Gaussian hills with a height of 0.1 kJ/mol and width of 0.02 nm were added to the biasing potential every 1 ps. The final energy was constructed finally as the negative of the biasing potential. We stopped depositing the Gaussians as soon as the stalk formation took place, to avoid overfilling the energy well.

### Analysis MD data

The simulation data were analyzed using GROMACS tools, as well as in-house codes in Python, incorporating the 2.3.0 version of MD Analysis[77,78]. The 1.9.3. version of VMD[79] and the 1.7.dev2023 version of ChimeraX[62,63] were used to visualize the structures and trajectories, as well as to prepare the snapshots and the movies.

The pore volume as displayed in Fig. 3 was created by moleonline[80]. To calculate the pore profiles of the proteins, the 2.2.004 version of the HOLE program[81] was used. For all simulations, the pore profile was calculated every 10th ns, and the average and the standard deviation of the pore were calculated and then represented with a solid line and a shaded area, respectively.

To calculate the ion permeation events, upper and lower thresholds corresponding to the z-position of the Cα atoms of the S92 and G61 residues were considered, respectively, and then when an ion during the simulations had a continuous trajectory with the maximum z-position value higher than 5 Å above the upper threshold and the minimum z-position value lower than 5 Å below the lower threshold, the ions were considered as crossed and the trajectory of the ion was recorded.

To calculate the density of the ions along the xy-axis, the protein was considered at the center and the ion positions along the xy-axis were accordingly translated. The histograms were calculated for different slabs along the z-axis and averaged over the simulation time.

For the distance of the two adjacent helices, for all Cα atoms of one helix, the minimum distance to a Cα atom from the second helix is determined. The overall distance is given as the average value of these distances.

### Force Field

For all simulations we have used the version CHARMM36m of the CHARMM force field[82]. For Ca2+ the multi-site model has been used[83].

### Reporting summary

Further information on research design is available in the Nature Portfolio Reporting Summary linked to this article.

## Data availability

The hybrid cryo-EM maps of "prepore" and "pore" state have been deposited in the Electron Microscopy Data Bank (EMDB) under accession codes EMD-51494 and EMD-51495, respectively. Corresponding maps of focused refinements on individual regions of the complex and the consensus refinements are available under accession codes: EMD-51467 (Prepore,chainA,core); EMD-51465 (Prepore,chainA,arm); EMD-51468 (Prepore,chainB,core); EMD-51469 (Prepore,chainB,arm); EMD-51472 (Prepore,chainC,core); EMD-51473 (Prepore,chainC,arm); EMD-51474 (Prepore,chainD,core); EMD-51475 (Prepore,chainD,arm); EMD-51492 (Prepore, consensus refinement); EMD-51467 (Pore,chains A-D,core); EMD-51465 (Pore,chainA,arm); EMD-51465 (Pore,chainB,arm); EMD-51465 (Pore,chainC,arm); EMD-51465 (Pore,chainD,arm); EMD-51492 (Pore, consensus refinement). The cryo-EM dataset has been deposited to EMPIAR under accession codes EMPIAR-12308. The atomic coordinates of the corresponding models have been deposited in the Protein Data Bank (PDB) under accession codes 9GO9 (Prepore state) and 9GOA (Pore state). The MD input files, starting structures and final structures as representative configurations can be found under https://zenodo.org/records/12663805. Other data are available in the source data published alongside this manuscript. Source data are provided with this paper.

## Code availability

Custom code used to generate data described in the manuscript is available in the source data file published alongside this manuscript.

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

## Acknowledgements

This work was supported by the DFG (SFB 1348 to A.H. and C.G.). The cryo-EM data were collected at "Cryo-EM SoN", the cryo-EM infrastructure of the University of Münster, funded by the Deutsche Forschungsgemeinschaft (DFG, German Research Foundation)—project number 496113311. Work in the lab of C.G. by M.C. was supported by the Humboldt Research Fellowship for Postdoctoral Researchers (to M.C). We thank Dr. Neuhaus and Dr. Blanque (cryo-EM SoN, Center for

Soft Nanoscience) for their technical support. The cryo-EM dataset was processed at the Palma II HPC (DFG INST 211/667-1) of the University of Münster. The MD simulations have also been carried out on the Palma II cluster. We acknowledge technical support by the HPC team of the University of Münster.

## Author contributions

C.G. designed and managed the project. B.U.K. collected and processed cryo-EM data with contributions by M.C. at the initial stage of the project. B.U.K. analyzed cryo-EM data, performed AF2 predictions, built atomic models, and interpreted data with contributions by C.G. K.K.S. and M.C. performed liposome reconstitution experiments, negative stain EM screenings, and processed negative stain EM data. A.A. performed MD simulations. A.A. and A.H. analyzed MD data. B.U.K. and C.G. drafted the manuscript. B.U.K., A.A., A.H., and C.G. wrote the manuscript. All authors discussed the results, commented, and approved the manuscript.

## Funding

## Competing interests

The authors declare no competing interests.
