## [Peer Review File · Nature Communications]

Structural basis of α -latrotoxin transition to a cation selective poreReviewer #1 (Remarks to the Author):

In the present work, Klink et al. report the high-resolution cryo-EM structures of the α -latrotoxin (α -LTX) tetramer in both prepore and pore states. α -LTX is a neurotoxic component found in the venom of black widow spiders. The 130 kDa protein is characterized by an N-terminal helical region (CD), a central domain (HBD), a beta-sheet plug module (PD) and a C-terminal tail of 22 ankyrin repeats. The cryo-EM reconstructions reveal that the helices of the N-terminal domain (CD) undergo a large conformational change to self-assemble into an extended coiled-coil needle. Although the distal end of the tetrameric needle was not well resolved in the cryo-EM density, the authors employ AlphaFold2 predictions and molecular dynamics simulations to propose that the N-terminal helices of the protein could form a cation-permeable transmembrane pore. Overall, the structural work is generally solid and provides important insights into the mechanisms by which pore-forming toxins penetrate and render the plasma membrane permeable to certain ions. The only significant issue with the manuscript at present is that the authors should modulate some of the statements regarding the comparisons with other systems, as well as some of the conclusions extracted from the bioinformatic results (see below). This matters aside, the findings are of broad interest across a range of fields.

Specific points:

- The AlphaFold2 predictions and molecular dynamics simulations nicely support some of the current and previous observations and help to propose, for instance, different cation selectivity, entry/exit gates, and mechanisms of action. Some of the bioinformatic analyses, however, do not have direct experimental confirmations and, thus, the wording of some of the conclusions drawn from these analyses in Results and Discussion may be overly assertive and should be appropriately tempered.
- The comparative analyses with other systems are interesting and highlight crucial similarities and discrepancies. Some of the observations, however, are not completely accurate. Although it has been reported that, similarly to α -LTX, the Vip3 coiled coil can accommodate small ions, the internal pore of the needle is significantly hydrophobic and narrow and it was never clear whether it could permit ion passage through the whole structure (Núñez-Ramírez R, et al. Nat Commun. 2020). It was rather suggested that the first α -helix of Vip3 could, which is aliphatic and flexible in solution, could form an ion permeable channel in contact with the membrane (Núñez-Ramírez R, et al. Nat Commun. 2020), a model that has been further supported in later reports (e.g., Lázaro-Berenguer M, et al. Microb Biotechnol. 2022; Shao E, et al. Toxins. 2024). The results described here for α -latrotoxin, thus, align well and are consistent with the models proposed for Vip3.
- The secondary rank of *Bacillus thuringiensis* Vip toxins (e.g., Vip3A) should be capitalized.
- The structural work is robust and the figures and videos, which are beautiful and illustrative, help to easily follow the story. Notably, however, the particles adopt a marked preferred orientation on the EM grids and the authors have to collect several datasets different tilt angles. Interestingly, the angular distributions of the merged datasets (Supp Fig 1e and f) still show a prominent preferred orientation, even considering that the particles at 0 tilt will be more concentrated on that particular area. Do the different 2D and 3D classification steps get rid of a significant fraction of the tilted particles (perhaps due to an increased motion or worse CTF

estimation)? Did the authors try to use detergents or support films to try to alleviate this issue? Could the authors provide the phase-randomized FSC curves for the global reconstructions (i.e., the ones obtained before performing multi-body refinement)?

- Since both the prepore and pore configurations are mainly tetramers, why is the data processed using C1 symmetry? This would allow them to see some of the breathing movements that they describe in the manuscript, but sometimes processing symmetric structures without imposing symmetry can lead to the appearance of non-physiological symmetry breaks (e.g., reinforcing the alignment of the better-defined monomers of each molecule). What happens when the structures are processed with C4 symmetry?

- The EM reconstructions show that the helices of the CD domain rearrange into a coiled-coil configuration. Although it is not surprising to find that the tip of the extended needle is flexible, did the authors try different processing strategies to improve the density of the N-terminal region (e.g., particle subtraction + 3D classification)? It would be interesting to have an experimental validation of the TMD model predicted with AlphaFold with the truncated sequence of the protein.

- Although the cryo-EM maps show a wide resolution range (and some anisotropy), the authors can build atomic models based on previous results further supported by AlphaFold models. Overall, the atomic models are fine, but the PDB validations show some geometry parameters that need some work (i.e., rotamer outliers are 4 and 9%). The cryo-EM densities probably do not support those outlier percentages and the authors might want to consider fixing some of them and giving more weight to the geometry during refinement.

Reviewer #2 (Remarks to the Author):

Summary:

Klink et al. perform a structural biology investigation of the black widow spider latrotoxin (LTX), a pore-forming protein found in venom. The structural findings provide evidence for the oligomeric state, the transition of pre-pore to pore, and highlight key residues and the underlying structural basis for assembly and complex architecture. Further computational analyses provide hypotheses for ion selectivity and the authors propose a mechanism of ion-permeability as supported by MD simulations and structural observations.

Strengths:

- The structural biology and electron microscopy is technically sound and appears to be of high standard. I was impressed by the authors' efforts to reconstruct multiple conformations from an enormous (!!) dataset of ~74k micrographs, despite significant preferential orientation problems. While there remains some evidence of anisotropy, the density appears to be sufficient to model the overall domain re-arrangements (the authors did not provide maps or models). Nevertheless, the quality of the work is suitable for Nature Communications – however, I would caution the authors to be more conservative with the use of “high resolution”

as generally speaking the maps are 3-4 Å. Many groups have achieved significantly higher resolution (<2 Å) and the term “high resolution” is subjective.

Weaknesses:

- The broad statements about mechanism and ion transport are largely inferred by observations from two cryoEM reconstructions aided by MD simulations (and AlphaFold models). While these data are technically sound, I found the conclusions difficult to accept without some biological in vitro or in vivo supporting evidence. Nanlon SURFE2R or other electrophysiology could be carried out to test these hypotheses. Alternatively, cultured cell lines that are susceptible to LTX could be used to test activity. Overall, the analysis around ion permeability serves as an excellent hypothesis, however it requires further biological evidence.
- My understanding is the long channel in LTX is too small to allow Ca²⁺ ions to pass through – the authors should test this. Make mutations at the top of the channel and the bottom of the channel, do these affect Ca²⁺ permeability? E.g. D210/D219 vs D93/E38

General comments:

- I don't believe the title is a good reflection of the major findings of the study. The mechanism of pore formation remains largely unclear to me – what is the trigger for pore formation? How is receptor mediated engagement relayed to the CD domain to trigger insertion? Do ions travel through the CD region or via the full channel? I suggest re-phrasing the title to reflect the major accomplishments of resolving the pre-pore and pore state of LTX.
- The final statement of the abstract is quite broad and I don't think it's a good reflection of the major findings of the study. How will the prepore and pore structure provide a framework for therapeutics? What biotechnological applications? These should be addressed in the discussion or omitted.
- It seems that symmetry was not applied to solve the final complex. This seems sensible given that it appears there are significant deviations from any C₄ (pseudo)point group symmetry – however did the authors attempt to perform C₄ refinement followed by symmetry expansion (back to C₁) and then localised reconstruction of a single subunit? Alternatively, symmetry relaxation may also be beneficial. These methods may substantially improve the resolution of the final map, since flexibility appears to be a major factor / source of heterogeneity.
- Did the authors try detergent additives or continuous ultrathin carbon to circumvent the orientation problems? MotionCor3 has also demonstrated significantly improved performance on tilted data and led to improved reconstructions with less anisotropy – the authors may consider re-extracting their final particle stacks from MotionCor3 corrected movies and attempting a final refinement.
- The authors state “despite efforts, we did not succeed to induce pore formation in the presence of detergents” – this could be detailed in the methods, I find it quite surprising that detergent would inhibit pore formation. What about solubilisation and affinity capture from susceptible cell culture?
- Presumably the AR may function as receptor binding domains – binding may destabilize the 2x4H bundle enabling pore formation. Have the authors tried to analyse the LTX complex by ConSurf? This may provide clues as to the conserved interfaces on the AR domains that mediate receptor binding... Overall it remains unclear to me what the role of the AR domains is – other than oligomerisation and stabilisation of the pre-pore state. Maybe expand in the discussion.

- I found that I had to re-read many sentences to understand the author's meaning – I would respectfully recommend that the authors consider simplifying the text where possible, this will help make the article more accessible to the general audience of Nature Communications.
- The authors should consider simplifying the colour schemes of their figures. This is largely for clarity, e.g. fig 1a, colour a single subunit in the map and then show a single subunit separately as a larger panel with discrete domain colours. Colouring all AR domains seems unnecessary, and the figures become overly complex and difficult for a non-expert reader to decipher. Having such varied colours in each figure poses challenges to colour blind individuals. There is no need for background colour.
- The clash score of 13.6 and 15.8 seems very high – perhaps the models should be carefully inspected. It may benefit from some flexible refinement (e.g. ISOLDE or Amber).
- In the abstract: “synchronously rearrange” is stated – I assume this is related to the four helices of a single subunit (i.e. the CD region). This statement could be misunderstood as meaning that all four subunits simultaneously rearrange during pore formation, however there is no evidence that the complex forms a pore in a concerted manner (as opposed to each subunit undergoing conformational changes at slightly different time points and independently).

Minor comments:

- Define acronyms before their first use, “cryoEM” is not defined - I realise it is quite common, but it is good practice to define a term before its use.
- The AlphaFold model in Fig 2c/Supp 8a needs to show the pAE plots. Currently the authors show the pLDDT, however since this is a complex prediction, the pAE plots are needed to judge the quality of the predicted interface.
- Line 262: Eszett (β) is not equivalent to beta (β). Beta should be used.
- Lines 100-103: The phrasing of this sentence “Tetramerization does neither...” does not sound natural. Suggest “Neither large-scale conformational changes nor exposure of hydrophobic patches are required for tetramerization, indicating that state 1 corresponds to the prepore state of α -LTX”
- Line 105/107: “Stronger interactions” & “weaker interactions” – as far as I can tell there is no experimental evidence that underpins this statement. Are these judged by the structure alone?

It was a pleasure to see the significant progress you have made since your 2021 study and the first structures presented at Prato Pores years ago.

Charles Bayly-Jones

Reviewer #3 (Remarks to the Author):

Klink et al. present a structural model for the mechanism of membrane insertion and pore formation by the vertebrate-specific toxin α -latrotoxin (α -LTX) employed by the black widow spider. This model is based on new cryo-EM structures of the α -LTX tetramer in its prepore and pore states in combination with structural modeling. Using a clever series of molecular dynamics simulations, the authors first confirm the stability of the combined structure and then probe various hypotheses about the ion permeation mechanism.

The scientific results presented here are quite appealing and I very much enjoyed reading about

them. Overall, the paper is understandably written and the conclusions are well supported by the underlying structures and simulations. I have some concerns with details of the work that I am listing below. But I expect it will only take minor efforts to address them. Aside from these, I can highly recommend this manuscript for publication.

- Line 220: This paragraph mentions “the different Boltzmann factor” but does not describe how or why it changes.

- Line 253: The mere existence of trimers does not exclude that two dimers are needed to form a tetramer. A trimer could as well be a dead end in the assembly process. I would recommend to tone down this statement and focus on the structural evidence - which, I agree, supports the viability of monomer-by-monomer assembly (but would also allow two dimers to merge to a tetramer).

- Line 270: This sentence describes “the strong loss in free energy when slowly removing one monomer” but the free energy *increases* when removing the monomer. The subsequent conclusion (that the tetrameric state is more favorable) appears to be unaffected by this mix-up.

- Line 296: Change ”predictions further confirm” to “predictions suggest”

- Lines 351f: Since “the presence of receptors significantly increases the occurrence of pores” I am wondering whether the structure from this paper can also provide insights into how interactions with receptors could increase pore formation.

- Lines 396ff: The authors mention potential technological applications of their work. However, these are likely not obvious to a general audience. It would help the reader understand the practical importance of their work if they provided some more details about how they think their model could be used in these developments.

- Line 407f: In my experience, data available “upon request” are effectively unavailable, especially after a few years. It would help others to reproduce and build on the authors’ work if they publicly deposited the starting structures and parameters for the MD simulations (They can easily be uploaded to, e.g., GitHub, Zenodo etc.) - especially since the methods section is very vaguely written in respect to how simulations were prepared. Ideally, they would also provide a representative structure of the equilibrated systems and any non-trivial analysis code and settings.

We thank the reviewers for their helpful comments on our work and useful guidance in revising our paper. Below is a point-by-point response to all comments and a detailed description of all changes we have made to our manuscript after considering their suggestions.

Reviewer 1

In the present work, Klink et al. report the high-resolution cryo-EM structures of the α -latrotoxin (α -LTX) tetramer in both prepore and pore states. α -LTX is a neurotoxic component found in the venom of black widow spiders. The 130 kDa protein is characterized by an N-terminal helical region (CD), a central domain (HBD), a beta-sheet plug module (PD) and a C-terminal tail of 22 ankyrin repeats. The cryo-EM reconstructions reveal that the helices of the N-terminal domain (CD) undergo a large conformational change to self-assemble into an extended coiled-coil needle. Although the distal end of the tetrameric needle was not well resolved in the cryo-EM density, the authors employ AlphaFold2 predictions and molecular dynamics simulations to propose that the N-terminal helices of the protein could form a cation-permeable transmembrane pore. Overall, the structural work is generally solid and provides important insights into the mechanisms by which pore-forming toxins penetrate and render the plasma membrane permeable to certain ions. The only significant issue with the manuscript at present is that the authors should modulate some of the statements regarding the comparisons with other systems, as well as some of the conclusions extracted from the bioinformatic results (see below). This matters aside, the findings are of broad interest across a range of fields.

We thank the reviewer for her/his appreciation of our work.

1) The AlphaFold2 predictions and molecular dynamics simulations nicely support some of the current and previous observations and help to propose, for instance, different cation selectivity, entry/exit gates, and mechanisms of action. Some of the bioinformatic analyses, however, do not have direct experimental confirmations and, thus, the wording of some of the conclusions drawn from these analyses in Results and Discussion may be overly assertive and should be appropriately tempered.

Thank you for your constructive comment. Please consider that α -latrotoxin has been used for decades as a model protein to study synaptic vesicle exocytosis, and its electrophysiological properties are very well characterized. The prediction of the TMD and the MD simulations, which were carefully performed to challenge the prediction, are in full agreement with the electrophysiological and biochemical properties of the α -LTX channel. To name some of those, the pore is cation selective, impermeable to Cl^- , and blocked by La^{3+} . Additionally, Na^+ is transported faster than Ca^{2+} . The toxin abolishes the pore-forming activity by removing the conserved stabilizing disulfide bond of the channel. Based on the MD results and the prediction, we discussed the general features of the pore and emphasized that they are in perfect agreement with the wealth of available literature, which is carefully cited and discussed accordingly. We apologize if the wording might sound assertive in some sentences. We have carefully revised the respective section of the manuscript and highlighted the changes in yellow.

2) The comparative analyses with other systems are interesting and highlight crucial similarities and discrepancies. Some of the observations, however, are not completely accurate. Although it has been reported that, similarly to α -LTX, the Vip3 coiled coil can accommodate small ions, the internal pore of the needle is significantly hydrophobic and narrow and it was never clear whether it could permit ion passage through the whole structure (Núñez-Ramírez R, et al. Nat Commun. 2020). It was rather suggested that the first α -helix of Vip3 could, which is aliphatic and flexible in solution, could form an ion permeable channel in contact with the membrane (Núñez-Ramírez R, et al. Nat Commun. 2020), a model that has been further supported in later reports (e.g., Lázaro-Berenguer M, et al. Microb Biotechnol. 2022; Shao E, et al. Toxins. 2024). The results described here for α -latrotoxin, thus, align well and are consistent with the models proposed for Vip3.

We apologize for this and thank you very much for pointing it out. We have rephrased the relevant section in the main manuscript, we have updated the reference list to include these citations, and we have also updated Supplementary Figure 11, accordingly, to emphasize the consistency of the models proposed for Vip3A and LTX.

3) The secondary rank of *Bacillus thuringiensis* Vip toxins (e.g., Vip3A) should be capitalized.

Done

4) The structural work is robust and the figures and videos, which are beautiful and illustrative, help to easily follow the story. Notably, however, the particles adopt a marked preferred orientation on the EM grids and the authors have to collect several datasets different tilt angles. Interestingly, the angular distributions of the merged datasets (Supp Fig 1e and f) still show a prominent preferred orientation, even considering that the particles at 0 tilt will be more concentrated on that particular area. Do the different 2D and 3D classification steps get rid of a significant fraction of the tilted particles (perhaps due to an increased motion or worse CTF estimation)?

We thank the reviewer for their positive comments. We indeed observed an asymmetric exclusion of particles dependent on the tilt angle for the prepore state. This is somewhat expected due to factors such as worse SNR, CTF estimation, and the overlap of particles at high tilts, as also mentioned by the reviewer, which might affect alignments and 3D classifications. However, this particle loss is not detrimental. We would like to emphasize that the final dataset includes 221.273 high-tilt particles. This is a significant number of particles for a cryoEM dataset reaching this resolution range, thus, we do not consider the size of the final tilted subsets to be a limitation. We have extended Supplementary Table 1 with two additional columns showing the number of particles in each final particle stack for each tilt angle to provide a better overview.

Surprisingly, we did not observe this asymmetric particle loss for the pore state (see updated Supplementary table 1). We interpret this as follows: the particles in this state have a characteristic umbrella-like conformation, compared to the flat, windmill-like conformation of the prepore state. When tilting the stage, the characteristic long stalk of the pore state particles can be seen, providing an additional feature that apparently improves the accuracy of alignments and counteracts the imaging limitations at high tilts mentioned above, compared to the particles of the prepore state.

To further validate the contribution of high-tilt and low-tilt subsets to the final reconstruction of the prepore, we performed additional refinements and compared the reconstruction from all particles in the final particle stack to those from the low tilt (0° - 21°) and high tilt (30° - 60°) subsets of the prepore. The amounts are approximately equal as the ratio of low to high tilt is around one. The nominal resolution according to gold standard FSC reached 3.12 \AA for all data, 4.16 \AA for the low tilt subsets, and 3.69 \AA for the high tilt subsets. For direct comparison, all maps were sharpened to 3.2 \AA and displayed at the same threshold. In **Reb. Figure 1**, we show a rather challenging segment of the map. Both low and high tilt data contribute significantly to the final map quality, and combining them was crucial for achieving more accurate "modellability" of the cryo-EM maps.

Reb. Figure 1 Representative segment of the 3D reconstruction from all particles of the final prepore dataset (left panel), from the particles of the low-tilt subsets (0 - 21° ; middle panel) and the particles of the high-tilt subsets (30 - 60° ; right panel), together with the respective 3D FSC plots.

5) Did the authors try to use detergents or support films to try to alleviate this issue? Could the authors provide the phase-randomized FSC curves for the global reconstructions (i.e., the ones obtained before performing multi-body refinement)?

The reviewer can be assured that we have explored all possible sample optimization options to obtain more views. For example, we tested grids with graphene oxide and carbon support (**Rebuttal Figure 2a-c**). We also experimented with alternative glow discharging methods using a prototype glow discharger to render the surface of grids positive or negative.

Additionally, we used a wide range of detergents and screened various concentrations, including octyl maltoside, Cymal-5, CHAPS, and DDM. Despite these efforts, we were unable to overcome the preferred orientation issue, as we consistently observed the characteristic top views in numerous cryoEM screening sessions. Interestingly, adding detergents led in several cases to the dissociation of tetrameric particles into monomers (Rebuttal Figure 2d-f). Thus, our only option to obtain the structures presented in this study was to undertake this significant effort and collect tilted data. It took 24 days of data collection on the most advanced and fastest cryoTEM to collect ~90,000 tilted images (we recently installed the new cryoTEM centre at our university) to solve the high resolution structures of both states of the toxin.

Reb. Figure 2 a-LTX particles on different grid types. a) open hole grids (used for dataset collection) b) additional amorphous carbon support (2 nm) c) additional graphene support. (For improved contrast, images in panels b and c were collected using the Volta Phase Plate. Note that the orientation of the particles remains unchanged across all grid types. d) buffer with 0.026% DDM; White boxes show two tetramers with an unchanged orientation. e) 0.05% DDM incubated over night at room temperature; the sample became very heterogenous and shows mostly dissociated latrotoxin monomers. Note that the micrographs in a-c, d, and e-f were collected at different magnifications. Scale bars, 50nm.

The phase-randomized FSCs for the global reconstructions of the prepore and pore state (before performing multi-body refinement) are shown as a new panel g) in the revised Supplementary Fig. 1.

6) Since both the prepore and pore configurations are mainly tetramers, why is the data processed using C1 symmetry? This would allow them to see some of the breathing movements that they describe in the manuscript, but sometimes processing symmetric

structures without imposing symmetry can lead to the appearance of non-physiological symmetry breaks (e.g., reinforcing the alignment of the better-defined monomers of each molecule). What happens when the structures are processed with C4 symmetry?

We thoroughly tested the use of C4 symmetry, including processing the data from scratch, and found that up to 5 Å resolution, the resulting maps were at first glance indeed similar to those obtained with C1 refinements. However, in the final processing steps, the maps generated with C4 symmetry showed slightly reduced resolution. Most importantly, direct comparison revealed that important high-resolution features, particularly in the central core of the volume, where the “breathing” motions in C1 reconstructions are observed, were smeared out (**Rebuttal Figure 3**). Importantly, the level of detail there did not match the reported resolution. Thus, we conclude that the tetramers are pseudo-symmetric, and at least below 5 Å, C1 symmetry is definitely the correct choice.

Reb. Figure 3 Side-by-side comparison of a representative region of the final density map of the prepore state, processed with C1 symmetry (left panel) and C4 symmetry (right panel), along with the corresponding FSCs.

7) The EM reconstructions show that the helices of the CD domain rearrange into a coiled-coil configuration. Although it is not surprising to find that the tip of the extended needle is flexible, did the authors tried different processing strategies to improve the density of the N-terminal region (e.g., particle subtraction + 3D classification)? It would be interesting to have an experimental validation of the TMD model predicted with alphafold with the truncated sequence of the protein.

The reviewer can be assured that we meticulously performed extensive focused 3D classifications to resolve the tip of the needle, but these efforts were unsuccessful. Even with strong filtering, no additional density could be found, which is often the case for flexible domains. Unlike the Vip3A structure, where long visible stalks were already seen in 2D classifications, our particles exhibit a strong preferred orientation, aligning to the air-water interface. We assume that the hydrophobic tip of the needle, once exposed, clashes with the air-water interface, preventing its resolution in the final 3D reconstruction. Additionally, attempts to stabilize the TMD using DDM failed, as the tetramers dissociated into monomers (**Rebuttal Figure 2**). We observed the full-length pore only after reconstituting the complex in a near-native lipid environment (liposomes; see main Figure 2). However, without receptors, the reconstitution efficiency is extremely low, precluding further high-resolution cryoEM studies of the complex in liposomes and/or nanodiscs at this stage.

8) Although the cryo-EM maps show a wide resolution range (and some anisotropy), the authors can build atomic models based on previous results further supported by AlphaFold models. Overall, the atomic models are fine, but the PDB validations show some geometry parameters that need some work (i.e., rotamer outliers are 4 and 9%). The cryo-EM densities probably do not support those outlier percentages and the authors might want to consider fixing some of them and giving more weight to the geometry during refinement.

We thank the reviewer for this comment. There are gradients of resolution in the density maps, and apparently, we have not focused enough on improving the models in lower resolution regions/domains (e.g., the ARs). We apologize for this and we have now improved the models, which is reflected in the significantly improved statistics. Please find a summary of the old vs new statistics in Rebuttal Table 1. The new values are also updated in Supplementary table 1. The rotamer outliers are reduced to 0.1% and 0.2%, respectively. We also truncated the C-terminal residues of the pore state (residues 975-1195, AR-tail) to stubs, as the density does not support side-chain placement.

Atomic model composition	Prepore (old)	Prepore (new)	Pore (old)	Pore (new)
Chains	4	4	4	4
Symmetry imposed	C1	C1	C1	C1
Non-hydrogen (protein) atoms	36,980	36,980	34,612	31,628
Residues	4,700	4,700	4,364	4320
particle substack	442,105	442,105	70,971	70,971
Ligand atoms	-	-	-	-
Refinement (Phenix)				
RMSD bond	0.005	0.002	0.007	0.004
RMSD angle	0.753	0.571	0.847	0.709
Model to map fit, CC mask	0.76 (0.75)	0.72 (0.73)	0.67 (0.66)	0.61 (0.60)
Model to map fit, CC box	0.60 (0.87)	0.57 (0.86)	0.71 (0.76)	0.67 (0.77)
Resolution (FSC@0.143, Å)	2.71 (3.15)	2.71 (3.15)	3.61 (3.69)	3.61 (3.69)
B-factor (mean, Å ²)	117.45	116.04	121.68	89.73
Validation				
Clashscore	13.64	8.75	15.79	9.71
Ramachandran outliers (%)	0	0.02	0.07	0.02
Ramachandran favoured (%)	95.89	96.12	96.05	96.29

Molprobity score	2.48	1.73	1.97	1.76
EMRinger score	1.50 (1.38)	1.46 (1.18)	1.41 (0.57)	1.40 (0.84)

Rebuttal table 1: Comparison of cryo-EM data collection and refinement statistics of α -LTX before and after model optimization with more weight on geometry. The refinement in Phenix was done against hybrid maps obtained by Phenix.combine_focussed_maps after multi-body refinement (Supplementary figure 2). The presented refinement statistics are for the hybrid maps (resolution is the reported resolution from multi-body refinement), and in brackets for the same model against the original, not locally refined nor combined maps after 3D-refinement and sharpening in RELION4.

Reviewer 2

Klink et al. perform a structural biology investigation of the black widow spider latrotoxin (LTX), a pore-forming protein found in venom. The structural findings provide evidence for the oligomeric state, the transition of pre-pore to pore, and highlight key residues and the underlying structural basis for assembly and complex architecture. Further computational analyses provide hypotheses for ion selectivity and the authors propose a mechanism of ion-permeability as supported by MD simulations and structural observations.

Strengths:

1) The structural biology and electron microscopy is technically sound and appears to be of high standard. I was impressed by the authors efforts to reconstruct multiple conformations from an enormous (!!) dataset of ~74k micrographs, despite significant preferential orientation problems. While there remains some evidence of anisotropy, the density appears to be sufficient to model the overall domain re-arrangements (the authors did not provide maps or models).

Nevertheless, the quality of the work is suitable for Nature Communications – however, I would caution the authors to be more conservative with the use of “high resolution” as generally speaking the maps are 3-4 Å. Many groups have achieved significantly higher resolution (<2 Å) and the term “high resolution” is subjective.

Thank you very much for the appreciation of our efforts.

With “*high resolution*” we rather wanted to emphasize that the resolution in the major part of the structure allows to confidently build an atomic model, which was not possible with earlier structural work on α -LTX (see Orlova et al. 2000). We agree and rephrased the wording throughout the manuscript according to your suggestion.

Maps and models were not requested during the review process, but we would be happy to provide them upon an editorial/reviewer request. Maps and models will be released in the EMDB and PDB upon acceptance of the manuscript and in addition we will submit all raw data and coordinates to EMPIAR. Open access to data enables independent validation, reuse, and integration of structural information. We strongly support the EMPIAR initiative and due to the clear benefits for our field, we hope that EMPIAR depositions will become mandatory in the future.

Weaknesses:

2) The broad statements about mechanism and ion transport are largely inferred by observations from two cryoEM reconstructions aided by MD simulations (and AlphaFold models). While these data are technically sound, I found the conclusions difficult to accept without some biological in vitro or in vivo supporting evidence. Nanlon SURFE2R or other electrophysiology could be carried out to test these hypotheses. Alternatively, cultured cell lines that are susceptible to LTX could be used to test activity. Overall, the analysis around ion permeability serves as an excellent hypothesis, however it requires further biological evidence.

Thank you for your constructive comment. The reviewer likely refers to the *in silico* section of our manuscript, specifically the transmembrane domain (TMD), as the other critical aspects of the pre-pore-to-pore transition are based on experimental structures.

The N-terminal 336 residues of the 4700aa tetrameric mature toxin forming the TMD in the pore state, are not resolved in the cryoEM structure of the pore. However, this segment is resolved in the pre-pore state, revealing fragmented helices stabilized by a conserved disulfide bond. The predictions indicate that in the pore state, these fragmented helices straighten to form a long helical pair, still stabilized by the conserved disulfide bond, with the expected length to span the membrane bilayer. The resulting tetrameric transmembrane channel displays all typical features of a PFT TMD, including a negatively charged gate just above the upper membrane leaflet. This AlphaFold2 prediction was consistent across all members of the latrotoxin family, which are expected to have a conserved mechanism of channel formation.

We agree with the reviewer that the TMD model remains a prediction. Although it appears convincing, it certainly requires further validation. To address this, we have therefore dedicated significant effort to reconstituting the complex in liposomes and performing extensive molecular dynamics (MD) simulations to support our findings.

The negative stain EM experiments in liposomes confirm that the helices of this domain must rearrange and enter the membrane to form the TMD, as the rest of the structure is resolved and positioned directly above the membrane. Furthermore, the MD simulations, carefully conducted to challenge the TMD prediction, fully align with the electrophysiological and biochemical properties of the native α -LTX channel.

We want to emphasize that α -latrotoxin has been a model protein for studying neurotransmitter release for decades due to its high biomedical relevance. Its functionality has been extensively investigated, with numerous studies analyzing all aspects of ion transport.

Therefore, we thank the reviewer for appreciating our *in silico* work and considering the resulting hypothesis excellent. However, we kindly disagree with the assertion that there is no biological *in vivo* or *in vitro* evidence supporting our overall conclusions.

Characterization of the native toxin's electrophysiological properties began already in the 1970s. The pore is cation-selective, impermeable to Cl^- , and blocked by La^{3+} , with Na^+ transported faster than Ca^{2+} . Notably, the toxin's channel activity is completely abolished when the conserved stabilizing disulfide bond, which according to the prediction holds the helical pairs of the TMD bundle together, is removed.

Thus, regarding the electrophysiological properties of native α -latrotoxin, the TMD model has been rigorously challenged by MD simulations in presence of ions and has passed all tests. We believe our study exemplifies the careful integration of data from various state-of-the-art methods, which together converge on a channel architecture that fully aligns with the extensive biochemical and electrophysiological data available. Hence, it is unnecessary to repeat these well-conducted electrophysiology studies on the native toxin to re-confirm the prediction and the MD results.

We agree however with the reviewer that our study now provides for the first time significant insights to guide more specific mutational functional studies, which would indeed be of very high interest. However, we want to emphasize that here, we studied the native toxin purified from the venom. Due to biosafety reasons, we are unfortunately currently not able/allowed to perform such mutational experiments in our laboratory with the human pathogenic α -latrotoxin. Recombinant expression of the insect-specific homologue toxins is however established in our laboratory; those are expected to have a conserved mechanism of action but unfortunately, distinct electrophysiological properties.

According to the suggestion of the reviewer, we nevertheless initiated such efforts on the insect-specific homologues and used the combination of the Nanion Vesicle Prep and the Nanion Port-A-Patch to try to generate more specific structure guided functional data on the proteins and respective variants. Vesicles were prepared using DPhPC as lipid and cholesterol under the conditions suggested by Nanion. Small diameter chips were applied. The vesicles were then diluted and giga-seal formation was achieved, but with a low success rate. Unfortunately, we have not been able to apply the wt protein and still maintain the seal. At this stage, this could be due to technical or handling problems. We therefore contacted Nanion, but even with their expertise these problems could not be solved. Nanion suggested the Orbit mini as a more suitable alternative instrument for functional studies of reconstituted ion channels. We are in the process of obtaining the Nanion Orbit mini to be able to complement future studies of the insect-specific latrotoxins with functional recordings.

We also already invested significant efforts to provide an experimental structure of the TMD (such as detergent screenings, see response 4 to Reviewer 1 and response 9 to Reviewer 2) and optimizing reconstitution into liposomes or nanodiscs, but the reconstitution efficiency is too low, in absence of receptors, currently precluding any high resolution studies. To achieve this goal, we would need to express and purify the full-length receptors (the adhesion GPCR latrophilin or the presynaptic receptor neurexin) and possibly reconstitute them in liposomes or nanodiscs to increase the efficiency of membrane insertion by the toxin in a close-to-native environment. Based on previous studies, binding to receptors is expected to significantly improve membrane insertion efficiency. Another alternative is to express and purify the TMD alone and, if it assembles successfully, perform either crystallography or NMR, as the TMD consists only of four pairs of short α -helices. Such results cannot be accomplished within the scope of this manuscript. Solving the toxin/receptor structures in the native environment or the TMD alone will require at least a couple of years and is beyond the scope of this study.

We also want to emphasize that elucidating the pore formation mechanism for PFT systems of similar complexity has required a series of publications. For example, the pore-forming Tc-toxin complex was detailed over multiple studies (Gatsogiannis et al., 2013, Nature; Meusch, Gatsogiannis et al., 2014, Nature; Gatsogiannis et al., 2016, NSMB; Gatsogiannis et al., 2018, Nature).

We apologize that we cannot provide further functional data and an experimental structure of the TMD at this stage and we also apologize if the wording might have been too broad. We therefore have now carefully considered the comments of the reviewer and rephrased the manuscript accordingly to emphasize current limitations of our study.

3) My understanding is the long channel in LTX is too small to allow Ca^{2+} ions to pass through – the authors should test this. Make mutations at the top of the channel and the bottom of the channel, do these affect Ca^{2+} permeability? E.g. D210/D219 vs D93/E38

We appreciate the reviewer's constructive comment. Regarding mutational studies, please note our response to the previous comment. The cryoEM structure and MD simulations of the stalk in presence of ions clearly confirm that the diameter of the coiled-coil stalk is too narrow for Ca^{2+} ions to pass through. The radius of the tetrameric coiled-coil is consistently less than 2 Å. Furthermore, trapped cations in the cavity directly above the stalk were never released during the simulations and even when a pulling force was applied to push the bound Ca^{2+} ion into the stalk, the cation quickly exited sideways instead (Supplementary figure 10b). No Na^+ was observed to enter the stalk either. Furthermore, the rigidity and numerous hydrogen bonds within the coiled-coil, even when subjected to pulling forces aimed at disassembly or destabilization, prevent any structural re-arrangements that could alter its diameter (Supplementary Figure 12). Interestingly, recent functional studies published this year on Vip3A, which also features a tetrameric coiled-coil of similar diameter and a helical terminal TMD, demonstrate that also in this PFT the coiled-coil is not involved in ion transport, but exclusively the TMD.

General comments:

4) I don't believe the title is a good reflection of the major findings of the study. The mechanism of pore formation remains largely unclear to me – what is the trigger for pore formation? How is receptor mediated engagement relayed to the CD domain to trigger insertion? Do ions travel through the CD region or via the full channel? I suggest re-phrasing the title to reflect the major accomplishments of resolving the pre-pore and pore state of LTX.

Our data clearly suggest, as we thoroughly explain in the manuscript (see also Comment 3), that ions travel exclusively through the TMD, entering a gate positioned directly above the upper membrane leaflet. The funnel and the stalk are not involved in ion transport. Regarding the receptor-mediated triggering of pore formation, we discuss this point in response to Comment 5 of Reviewer 3. We thank you for the suggestion. We agree to change the title of the manuscript, as the action of α -latrotoxin at the presynaptic membrane involves in addition interaction with receptors, which is not in focus of the present manuscript.

The new title is:

Structural basis of α -latrotoxin transition to a cation selective pore

5) The final statement of the abstract is quite broad and I don't think it's a good reflection of the major findings of the study. How will the prepore and pore structure provide a framework for therapeutics? What biotechnological applications? These should be addressed in the discussion or omitted.

Please refer to our response to Comment 6 from Reviewer 3. Nonetheless, we thank you for your suggestion and taking this in account, we have rewritten the final statements of the discussion to focus on highlighting outstanding questions about the role of receptor engagement in toxin action and specificity, rather than emphasizing potential applications.

6) It seems that symmetry was not applied to solve the final complex. This seems sensible given that it appears there are significant deviations from any C4 (pseudo)point group symmetry – however did the authors attempt to perform C4 refinement followed by symmetry expansion (back to C1) and then localized reconstruction of a single subunit? Alternatively, symmetry relaxation may also be beneficial. These methods may substantially improve the resolution of the final map, since flexibility appears to be a major factor / source of heterogeneity.

Indeed, we also believe that the limiting factor for map resolution is remaining heterogeneity due to the flexible nature of the complex. We did utilize symmetry relaxation to C4 as implemented in Relion to resolve misalignment of pseudo-symmetric particles. We clarified in the Methods section that C4 relaxation was used.

We did also perform localized reconstructions on single subunits/independently moving parts of the complex as part of the multibody refinement (Multibody refinement does signal subtraction on signal from all except one of the used input masks that define the individual bodies, and then does local refinement on the subtracted particle stack). Those locally refined maps were then recombined using Phenix “combine focused maps”. For the prepore state, we additionally split each subunit in two independently moving N-and C-terminal parts to account for intramolecular flexibility within each monomer. Since the four N-terminal domains are tightly interacting in the coiled-coil stalk of the pore state and therefore cannot be split in multiple bodies, only a single body was used for all four N-termini, but each C-terminal ARD of the pore was processed as an individual body. These locally refined reconstructions were indeed crucial to improve the resolution, in particular in the ARD domains.

7) Did the authors try detergent additives or continuous ultrathin carbon to circumvent the orientation problems?

Please see our response to Comment 5 of Reviewer 1.

8) MotionCor3 has also demonstrated significantly improved performance on tilted data and led to improved reconstructions with less anisotropy – the authors may consider re-extracting their final particle stacks from MotionCor3 corrected movies and attempting a final refinement.

Thank you for the hint. We will favor MotionCor3 for future projects, especially with tilted data. Unfortunately, when using MotionCor3 with our current dataset (collected with the Falcon 4i detector in .eer format), we observed a significantly worse gain correction compared to Relion4's internal MotionCor2-like implementation. Despite several attempts, we have been unable to resolve this issue. Since we already applied single-particle motion correction after the initial patch correction by MotionCor2, which only modestly improved resolution, we might expect only minor improvements from reapplying patch correction in MotionCor3.

9) The authors state “despite efforts, we did not succeed to induce pore formation in the presence of detergents” – this could be detailed in the methods, I find it quite surprising that detergent would inhibit pore formation. What about solubilization and affinity capture from susceptible cell culture?

We apologize that we did not elaborate this in greater detail. Our intention was not to suggest an inhibition of pore formation. Instead, we aimed to convey that while our liposome experiments produced a very small but detectable and analyzable amount of pore complexes with intact TMD, we did not observe intact TMDs in the detergent tests. In our detergent tests (**Rebuttal Figure 2d-f**), we rather observed a significant tendency towards dissociation. Under mild conditions (low detergent concentrations with short incubation times), complexes tended to fall apart, and the few remaining intact tetramers exhibited the same preferred orientation issues as before. Under harsher conditions (e.g., 0.05% DDM with overnight incubation at room temperature), the particles nearly completely dissociated into monomers, and “healthy” tetrameric assemblies were no longer observed. Incorporation into liposomes prevented these detrimental effects on the complex and additionally prevent the TMD from contacting the air-water interface. The extremely low reconstitution efficiency prevents however further high-resolution studies with this sample. We have clarified the relevant paragraph.

The suggestion to utilize affinity capture from susceptible cell culture would require establishing recombinantly expressed α -LTX, which is, as explained above, currently not possible. Additionally, given that detergents cause particle dissociation, we believe solubilizing the complex from cell culture might not be a feasible approach.

9) Presumably the AR may function as receptor binding domains – binding may destabilize the 2x4H bundle enabling pore formation. Have the authors tried to analyse the LTX complex by ConSurf? This may provide clues as to the conserved interfaces on the AR domains that mediate receptor binding... Overall it remains unclear to me what the role of the AR domains is – other than oligomerisation and stabilisation of the pre-pore state. Maybe expand in the discussion.

As suggested, we analyzed both our prepore and pore structures with the ConSurf tool (<https://consurf.tau.ac.il>, (Yariv et al. 2023)), see **Rebuttal Figure 4** for the analysis of the prepore model). To summarize the findings, regions of high conservation are present throughout the structure, with the tendency of structure-stabilizing regions like the stalk, the central helix of the HBD, the central beta-sheets of the PD and the inner “backbone” helices of the ARD being more conserved. Notably, the two cysteine residues of the TMD (C34, C91) are among the most conserved residues. There is no obvious single hotspot of low conservation, but consistent with the notion that the receptor binding occurs around the loop α 41 in α -LTX, this loop in particular and surrounding ARD repeats in addition present a lower conservation than the average rest of the structure. This could be due to the different needs to adopt to different host receptors that bind within this region of the ARD. This was already discussed in the manuscript (please see supplementary figure 15).

Reb. Figure 4 Analysis of sequence conservation of the α -LTX pore state by ConSurf. Regions with high conservation are colored in pink, and regions with high variability in cyan. Regions with insufficient data are colored in yellow (An example for a region that contains insufficient sequence data from other toxins is the loop containing helix α 41 in α -LTX, which is replaced by differently positioned loops in other latrotoxins; compare supplementary figure 15).

We believe that in addition to providing an oligomerization interface (LTX oligomerizes via PD-ARD interactions), the ARD has at least a binary role: first, to protect the N-terminal helices in the monomeric state, preventing toxic effects and/or protein degradation within the spider; and second, to provide the receptor-binding interface, properly positioning the toxin for membrane insertion. Whether receptor binding might also initiate events that support the rearrangement of the CD and stalk formation remains unclear. This is an interesting point raised by the reviewer. However, since we currently lack receptor-toxin structures and the precise receptor-binding site remains unidentified, further discussion would be speculative. This issue must be validated based on future receptor/latrotoxin complex structures. We address this in the discussion of the revised manuscript.

10) I found that I had to re-read many sentences to understand the author's meaning – I would respectfully recommend that the authors consider simplifying the text where possible, this will help make the article more accessible to the general audience of Nature Communications.

We carefully revised the paper with clarity in mind.

11) The authors should consider simplifying the colour schemes of their figures. This is largely for clarity, e.g. fig 1a, colour a single subunit in the map and then show a single subunit separately as a larger panel with discrete domain colours. Colouring all AR domains seems unnecessary, and the figures become overly complex and difficult for a non-expert reader to decipher. Having such varied colours in each figure poses challenges to colour blind individuals. There is no need for background colour.

We attempted to implement this suggestion but were not satisfied with the result. We maintain a consistent color code throughout the manuscript, which we believe is crucial for clarity and guiding the reader. Furthermore, we used background colors to highlight that the upper and lower rows in Figure 2b represent two distinct states, which may not be easily discernible at first glance. Moreover, other reviewers have praised the aesthetic quality of our figures and videos in the manuscript. Therefore, we have chosen to keep the figures unmodified. To enhance accessibility for color-blind individuals, we have already adjusted the color intensities significantly, ensuring that most details remain distinguishable even in grayscale versions of the figures.

12) The clash score of 13.6 and 15.8 seems very high – perhaps the models should be carefully inspected. It may benefit from some flexible refinement (e.g. ISOLDE or Amber).

We apologize for this and we have now improved the models, which is reflected in the significantly improved statistics (see Rebuttal table 1). We updated the Supplementary table 1 in the manuscript.

13) In the abstract: “synchronously rearrange” is stated – I assume this is related to the four helices of a single subunit (i.e. the CD region). This statement could be misunderstood as meaning that all four subunits simultaneously rearrange during pore formation, however there is no evidence that the complex forms a pore in a concerted manner (as opposed to each subunit undergoing conformational changes at slightly different time points and independently).

Thank you for pointing this out. Indeed, we did not mean to imply that all four subunits necessarily rearrange simultaneously. We nevertheless believe that the rearrangement could be somewhat cooperative, because it involves the formation of a stable tetrameric coiled-coil stalk. Independent rearrangement of individual subunits might result in a very unstable intermediate state. However, we agree with the reviewer that without direct evidence, it is important to avoid such implications. Therefore, we have modified the phrasing in the abstract to avoid any confusion or overstatements:

“Four distinct helical bundles rearrange and together form a highly stable, 15 nm long, cation-impermeable coiled-coil stalk.”

Minor comments:

14) Define acronyms before their first use, “cryoEM” is not defined - I realise it is quite common, but it is good practice to define a term before its use.

Done

15) The AlphaFold model in Fig 2c/Supp 8a needs to show the pAE plots. Currently the authors show the pLDDT, however since this is a complex prediction, the pAE plots are needed to judge the quality of the predicted interface.

Thank you for this suggestion. We included the pAE plots in Supplementary figure 8a.

16) Line 262: Eszett (β) is not equivalent to beta (β). Beta should be used.

Done

17) Lines 100-103: The phrasing of this sentence “Tetramerization does neither...” does not sound natural. Suggest “Neither large-scale conformational changes nor exposure of hydrophobic patches are required for tetramerization, indicating that state 1 corresponds to the prepore state of α -LTX”

Thank you for your suggestion. We have rephrased this sentence in the revised manuscript accordingly.

18) Line 105/107: “Stronger interactions” & “weaker interactions” – as far as I can tell there is no experimental evidence that underpins this statement. Are these judged by the structure alone?

These lines were indeed missing a link to Supplementary Table 4, in which we analyzed the binding energies of different interfaces of our final structural models using the PISA server. We apologize for this. A reference to Supplementary Table 4 is now included.

It was a pleasure to see the significant progress you have made since your 2021 study and the first structures presented at Prato Pores years ago.

Charles Bayly-Jones

Dear Charles,

Thank you for your constructive comments, which greatly improved our manuscript. We were disappointed to hear that the Prato Pores conference will not take place this year but look very much forward to future exchanges.

Reviewer 3

Klink et al. present a structural model for the mechanism of membrane insertion and pore formation by the vertebrate-specific toxin α -latrotoxin (α -LTX) employed by the black widow spider. This model is based on new cryo-EM structures of the α -LTX tetramer in its prepore and pore states in combination with structural modeling. Using a clever series of molecular dynamics simulations, the authors first confirm the stability of the combined structure and then probe various hypotheses about the ion permeation mechanism.

The scientific results presented here are quite appealing and I very much enjoyed reading about them. Overall, the paper is understandably written and the conclusions are well supported by the underlying structures and simulations. I have some concerns with details of the work that I am listing below. But I expect it will only take minor efforts to address them. Aside from these, I can highly recommend this manuscript for publication.

We thank the reviewer for the positive evaluation of our manuscript.

1) - Line 220: This paragraph mentions “the different Boltzmann factor” but does not describe how or why it changes.

We thank the reviewer for this comment. For the reason of clarity, we have substituted the Boltzmann factor and written:

First, for Ca^{2+} the resulting stronger driving force after applying the electric field yields higher selectivity.

2) - Line 253: The mere existence of trimers does not exclude that two dimers are needed to form a tetramer. A trimer could as well be a dead end in the assembly process. I would recommend to tone down this statement and focus on the structural evidence - which, I agree, supports the viability of monomer-by-monomer assembly (but would also allow two dimers to merge to a tetramer).

Our data on α LTX, along with our previous extensive study on other members of the latrotoxin family, α LCT and δ LIT (Chen et al., Nature Communications, 2021), which are homologues and expected to oligomerize in a similar manner to form the tetrameric prepore, do not support the notion that the toxin tetramer is exclusively formed upon the assembly of two dimers, as previously proposed (Ashton et al., Biochimie, 2000).

Our previous cryoEM study of α LCT and δ LIT provided models of the soluble α LCT monomer and δ LIT dimer (prepore) (Chen et al., Nature Communications, 2021), but we also showed 2D classes of trimers and tetramers for the insect specific member of the latrotoxin family. Representative 2D classes of δ LIT are shown in Rebuttal Figure 4 (Chen et al., Nature Communications, 2021), along with 2D classes of α LTX from this study.

Reb. Figure 4 Representative 2D class-averages of the oligomeric states of δ -LIT (Chen et al., 2021) and α -LTX (this study).

Thus, molecular models of the α LCT monomer, δ LIT dimer (Chen et al., Nature Communications, 2021), and the α LTX tetramer (prepore) (this study) are now available. Importantly, in the respective cryoEM datasets of both δ LIT and α LTX, the trimer is a significant particle population.

We want to emphasize that, based on the models of the α LCT monomer and δ LIT dimer, modelling the tetrameric arrangement (solved in this study) based on sequential assembly of monomers is straightforward (monomer \rightarrow dimer \rightarrow trimer \rightarrow tetramer), as this does not involve clashes or conformational changes. This is, however, also true for the assembly of the tetramer from two dimers (dimer + dimer \rightarrow tetramer).

Therefore, we agree with the reviewer: based on these data, although the existence of trimers strongly indicates a step-by-step assembly (the evidence for the trimer based on the 2D class averages is obviously compelling for its existence), this does not exclude the possibility that in addition, two dimers might also merge to form a tetramer. However, we still do not see any structural explanation that the formation of the tetramer solely depends on the merging of two dimers, as previously suggested.

We apologize for not elaborating on this in detail and have rephrased the respective section in the main manuscript.

3) - Line 270: This sentence describes “the strong loss in free energy when slowly removing one monomer” but the free energy **increases** when removing the monomer. The subsequent conclusion (that the tetrameric state is more favorable) appears to be unaffected by this mix-up.

For us the loss in free energy was equivalent to an “increase” in free energy. However, to avoid any misunderstanding we have reformulated this sentence as:

The stability of the coiled-coil is quantified by studying the increase in free energy when slowly removing ...

4) - Line 296: Change “predictions further confirm” to “predictions suggest”

Done

5) - Lines 351f: Since “the presence of receptors significantly increases the occurrence of pores” I am wondering whether the structure from this paper can also provide insights into how interactions with receptors could increase pore formation.

α LTX pore formation, as shown in our study, requires the assembly of the tetrameric prepore, which is also in consistence with previous functional data (Volynski et al. 2003; Khvotchev und Südhof 2000). *In vitro*, we can observe tetramers only at very high non-physiological concentrations used for cryoEM studies. At lower concentrations (we performed FPLC, negative stain EM for α -LTX, and additionally SEC-MALS for δ -LIT), the toxins are exclusively present as monomers. This strongly suggests that the oligomerization of the toxin to form the tetrameric prepore state, which is competent for membrane insertion, is concentration-dependent.

Similar to other α -helical pore-forming toxins, oligomerization prior to pore formation might be facilitated by binding to receptors. Receptor binding might enhance their local concentration by reducing their diffusion space from three dimensions in the extracellular medium to two dimensions on the cell surface.

Furthermore, our structures suggest, that the overall pore formation mechanism requires that the tetrameric prepore is properly oriented and in close proximity to the membrane, which might also be facilitated by binding to receptors. Interestingly, the length of the long stalk in the pore state matches the length of the receptors, which have an elongated shape, with the extracellular domain protruding far from the membrane.

It is however conceivable, that receptor binding might induce conformational changes that trigger oligomerization. We indeed observe conformational changes when we compare the α -LTX monomer extracted from the tetrameric prepore structure of α -LTX (this study) and δ -Lit dimer (Chen et al., 2021) with the soluble monomer of homologous α -LCT (Chen et al., 2021), prior to tetramerization, which is less extended.

In addition, receptor binding might additionally induce initial events that support the rearrangement of the CD and the formation of the stalk. This is an intriguing point raised by the reviewer. However, since we currently still lack receptor-toxin structures and the precise receptor-binding site remains unclear, we believe that any further discussion at the current stage of our research would be speculative and beyond the scope of this manuscript. Nevertheless, we revised the final statements of the manuscript to highlight the importance of such outstanding questions.

6) - Lines 396ff: The authors mention potential technological applications of their work. However, these are likely not obvious to a general audience. It would help the reader understand the practical importance of their work if they provided some more details about how they think their model could be used in these developments.

Latrotoxins hold significant biotechnological potential, including the development of improved anti-venoms, treatments for paralysis via reversible nerve terminal degeneration, mitigation of Botox-induced paralysis severity and duration, advanced molecular tools for studying exocytosis, novel fluorescent markers for adhesion receptors, and novel biopesticides. These applications have been widely discussed in the literature, and we have cited relevant studies.

A major limitation in this field has been the lack of mechanistic insights into α -LTX and its pore formation mechanism. Understanding the detailed structural and functional aspects of these toxins is crucial for such biotechnological advancements. For instance, designing novel toxin-antidotes and LTX-based biopesticides directly relies on structural insights. The importance of structural information in advancing this research should be clear to the broad audience, as

similar approaches have driven progress for biotechnological applications based on other bacterial and venom toxins. We anticipate that this will also be the case for latrotoxins. However, this foundational knowledge is an essential starting point, but must be now complemented by structural studies of toxin-receptor complexes and comparative analysis of different latrotoxin family members. This will elucidate phylum-specificity and enable the design of latrotoxin-inspired biopesticides with narrow specificity.

Therefore, we decided to revise the final statements of the manuscript.

7) - Line 407f: In my experience, data available “upon request” are effectively unavailable, especially after a few years. It would help others to reproduce and build on the authors’ work if they publicly deposited the starting structures and parameters for the MD simulations (They can easily be uploaded to, e.g., GitHub, Zenodo etc.) - especially since the methods section is very vaguely written in respect to how simulations were prepared. Ideally, they would also provide a representative structure of the equilibrated systems and any non-trivial analysis code and settings.

We appreciate this comment by the reviewer. The MD input files, starting and final structures as representative configurations can now be found under <https://zenodo.org/records/12663805>.

References

- Byrne, Matthew J.; Iadanza, Matthew G.; Perez, Marcos Arribas; Maskell, Daniel P.; George, Rachel M.; Hesketh, Emma L. et al. (2021): Cryo-EM structures of an insecticidal Bt toxin reveal its mechanism of action on the membrane. In: *Nature communications* 12 (1), S. 2791. DOI: 10.1038/s41467-021-23146-4.
- Khvotchev, Mikhail; Südhof, Thomas C. (2000): α -latrotoxin triggers transmitter release via direct insertion into the presynaptic plasma membrane. In: *The EMBO journal* 19 (13), S. 3250–3262. DOI: 10.1093/emboj/19.13.3250.
- Núñez-Ramírez, Rafael; Huesa, Juanjo; Bel, Yolanda; Ferré, Juan; Casino, Patricia; Arias-Palomo, Ernesto (2020): Molecular architecture and activation of the insecticidal protein Vip3Aa from *Bacillus thuringiensis*. In: *Nature communications* 11 (1), S. 3974. DOI: 10.1038/s41467-020-17758-5.
- Orlova, E. V.; Rahman, M. A.; Gowen, B.; Volynski, K. E.; Ashton, A. C.; Manser, C. et al. (2000): Structure of alpha-latrotoxin oligomers reveals that divalent cation-dependent tetramers form membrane pores. In: *Nature structural biology* 7 (1), S. 48–53. DOI: 10.1038/71247.
- Volynski, Kirill E.; Capogna, Marco; Ashton, Anthony C.; Thomson, Derek; Orlova, Elena V.; Manser, Catherine F. et al. (2003): Mutant alpha-latrotoxin (LTXN4C) does not form pores and causes secretion by receptor stimulation: this action does not require neurexins. In: *The Journal of biological chemistry* 278 (33), S. 31058–31066. DOI: 10.1074/jbc.M210395200.
- Yariv, Barak; Yariv, Elon; Kessel, Amit; Masrati, Gal; Chorin, Adi Ben; Martz, Eric et al. (2023): Using evolutionary data to make sense of macromolecules with a "face-lifted" ConSurf. In: *Protein science : a publication of the Protein Society* 32 (3), e4582. DOI: 10.1002/pro.4582.

Reviewer #1 (Remarks to the Author):

The authors have satisfactorily addressed my primary concerns in the revised manuscript. The modifications made in the new version, including the tempered statements regarding the comparisons with other systems and the conclusions derived from bioinformatic analyses, have significantly improved the clarity and accuracy of the paper. Notably, the efforts described by the authors to address the issue of preferred particle orientation on EM grids are commendable. In addition, the authors have also improved the atomic models by addressing the geometry parameters and outliers. The revised models now better reflect the cryo-EM densities and provide a more accurate representation of the structures.

Overall, the revised manuscript presents a solid and comprehensive study of α -latrotoxin, offering valuable insights into its structure and function. The structural and computational analyses are well-supported, and the findings are of broad interest to the field. I, therefore, consider this article suitable for publication.

Reviewer #2 (Remarks to the Author):

The authors have made substantial improvements to the clarity of the text and endeavoured to address many of my points. Considering the reasonable safety restrictions in place around the recombinant production of latrotoxin, I see now that additional *in vitro* validations of the proposed models and predictions would likely require substantial time and effort. Indeed, this seems well beyond the scope of this study. Furthermore, I acknowledge the high degree of difficulty that exists in establishing novel functional assays, especially in the context of membrane bilayers. Overall the advances and insights gained by these structural and *in silico* studies are nevertheless of great interest. It remains my opinion that further experimental validation of the proposed models, in order to conclusively link structural elements to specific functions/properties, would be of broad interest. Especially considering the availability of structural models reported herein. However, the current study is suitable for Nature Communications.

I have only minor typographical comments:

Line 360: "Our study provides [the] first"

Line 365: A period should be used in place of the colon after "(Figure 5)".

Reviewer #3 (Remarks to the Author):

My comments and suggestions were sufficiently addressed and I can now recommend this manuscript for publication. Very interesting work!

We thank the reviewers for their helpful comments on our work and useful guidance in revising our paper. Below is a point-by-point response to final comments.

Reviewer #1 (Remarks to the Author):

The authors have satisfactorily addressed my primary concerns in the revised manuscript. The modifications made in the new version, including the tempered statements regarding the comparisons with other systems and the conclusions derived from bioinformatic analyses, have significantly improved the clarity and accuracy of the paper. Notably, the efforts described by the authors to address the issue of preferred particle orientation on EM grids are commendable. In addition, the authors have also improved the atomic models by addressing the geometry parameters and outliers. The revised models now better reflect the cryo-EM densities and provide a more accurate representation of the structures.

Overall, the revised manuscript presents a solid and comprehensive study of α -latrotoxin, offering valuable insights into its structure and function. The structural and computational analyses are well-supported, and the findings are of broad interest to the field. I, therefore, consider this article suitable for publication.

We thank the reviewer for her/his appreciation of our work.

Reviewer #2 (Remarks to the Author):

The authors have made substantial improvements to the clarity of the text and endeavoured to address many of my points. Considering the reasonable safety restrictions in place around the recombinant production of latrotoxin, I see now that additional in vitro validations of the proposed models and predictions would likely require substantial time and effort. Indeed, this seems well beyond the scope of this study. Furthermore, I acknowledge the high degree of difficulty that exists in establishing novel functional assays, especially in the context of membrane bilayers. Overall the advances and insights gained by these structural and in silico studies are nevertheless of great interest. It remains my opinion that further experimental validation of the proposed models, in order to conclusively link structural elements to specific functions/properties, would be of broad interest. Especially considering the availability of structural models reported herein. However, the current study is suitable for Nature Communications.

Thank you for appreciating our work. We fully agree that further functional studies on α -latrotoxin would be of broad interest. Our structural study lays a strong foundation for this line of research, and we hope to offer such insights in a future study.

I have only minor typographical comments:

Line 360: "Our study provides [the] first"

Thank you for pointing out these typos. In line with the editors' recommendation to avoid terms like "new," "novel," or "first," we revised the sentence to: "Our study provides mechanistic insights into the process through which α -LTX integrates into presynaptic membranes..."

Line 365: A period should be used in place of the colon after "(Figure 5)".

Changed.

Reviewer #3 (Remarks to the Author):

My comments and suggestions were sufficiently addressed and I can now recommend this manuscript for publication. Very interesting work!

We thank the reviewer for her/his appreciation of our work.